# Cocoon: Robust Multi-Modal Perception with Uncertainty-Aware Sensor Fusion

**Minkyoung Cho[1], Yulong Cao[2], Jiachen Sun[1], Qingzhao Zhang[1],**
**Marco Pavone[2,3], Jeong Joon Park[1], Heng Yang[2,4], and Z. Morley Mao[1]**

[1]University of Michigan
[2]NVIDIA Research
[3]Stanford University
[4]Harvard University
`https://minkyoungcho.github.io/cocoon/`

## Abstract

An important paradigm in 3D object detection is the use of multiple modalities to enhance accuracy in both normal and challenging conditions, particularly for long-tail scenarios. To address this, recent studies have explored two directions of adaptive approaches: MoE-based adaptive fusion, which struggles with uncertainties arising from distinct object configurations, and late fusion for output-level adaptive fusion, which relies on separate detection pipelines and limits comprehensive understanding. In this work, we introduce Cocoon, an *object-* and *feature*-level uncertainty-aware fusion framework. The key innovation lies in uncertainty quantification for heterogeneous representations, enabling fair comparison across modalities through the introduction of a *feature aligner* and a learnable surrogate ground truth, termed *feature impression*. We also propose a training objective to ensure that their relationship provides a valid metric for uncertainty quantification. Cocoon consistently outperforms existing static and adaptive methods across both normal and challenging conditions, derived from natural and artificial corruptions. Furthermore, we demonstrate the validity and efficacy of our uncertainty metric across diverse datasets.

## 1 Introduction

In the rapidly evolving field of intelligent systems, accurate and robust 3D object detection is a fundamental and crucial functionality that significantly impacts subsequent decision-making and control modules. With the increasing importance of this field, there has been extensive research on vision-centric (Wang et al., 2022; Liu et al., 2022; 2023a; Wang et al., 2023) and LiDAR-based detection (Yan et al., 2018; Yin et al., 2021; Lang et al., 2019; Zhou et al., 2024), but both approaches have demonstrated weaknesses due to inherent characteristics in sensor data. For example, 2D data lacks geometric information, while 3D data lacks semantic information. To address these limitations, multi-modal approaches that integrate multi-sensory data are actively studied to enhance capabilities in areas such as autonomous vehicles, robotics, and augmented/virtual reality, setting a new standard for state-of-the-art performance in various perception tasks (Chen et al., 2023; Prakash et al., 2021; Bai et al., 2022; Li et al., 2022b; Yan et al., 2023; Li et al., 2022a; 2023; Xie et al., 2023).

The accuracy of multi-modal perception is significantly influenced by the operating context, such as time and the composition of surrounding objects (Yu et al., 2023; Dong et al., 2023). Given the diverse (often complementary) nature of different modalities, maximizing their utility across various objects and contexts remains challenging, particularly for long-tail objects (Feng et al., 2020; Curran et al., 2024). Existing research efforts to address this issue mainly fall into two categories: Mixture of Experts (MoE)-based approaches (Mees et al., 2016; Valada et al., 2017) and late adaptive fusion (Zhang et al., 2023; Lou et al., 2023; Lee & Kwon, 2019). Despite their contributions, existing methods often overlook the varying levels of informativeness among individual objects in the scene by applying weights uniformly across entire features (Fig.1a) or limiting comprehensive understanding/decoding by simply combining outputs from separately trained detectors (Fig.1b), both resulting

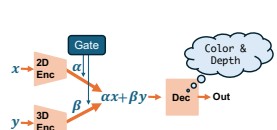 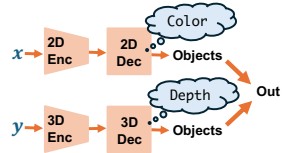 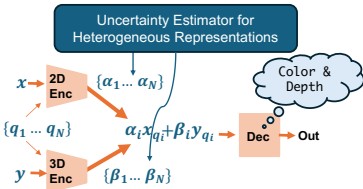

(a) MoE-Based Fusion: Whole feature fusion ignores object-level informativeness variation.

(b) Late Fusion: Each modality's detector is trained independently, limiting accuracy to that of the better-performing modality.

(c) Cocoon (Ours): $q_i$ indicates a query for a potential object.

Figure 1: Existing Fusion Methods[1] and Our Approach.

| | Object Distinction | Cross-Modal Feature Enrichment |
|---|:---:|:---:|
| MoE-Based Fusion (Mees et al., 2016; Valada et al., 2017) | ✗ | ✓ |
| Late Fusion (Lee & Kwon, 2019; Zhang et al., 2023; Lou et al., 2023) | ✓ | ✗ |
| Cocoon (Ours: object- and feature-level uncertainty-aware fusion) | ✓ | ✓ |

Table 1: Comparison with Prior Works

in sub-optimal performance. To our knowledge, none of these approaches address both object-level uncertainty variation and feature-level (i.e., intermediate) fusion needs, as highlighted in Table 1.

To address these challenges, we introduce Cocoon (Conformity-conscious sensor fusion), which tackles the aforementioned challenges by quantifying uncertainty at both the object and feature levels, and dynamically adjusting the weights of modality-specific features (Fig. 1c). Cocoon comprises four key components: a base object detector, feature alignment, uncertainty quantification, and adaptive fusion. For object-level adaptive fusion, Cocoon uses a query-based architecture as the base object detector, where a fixed number of queries are predetermined, each representing a potential object (Carion et al., 2020a; Chen et al., 2023; Bai et al., 2022). When multi-modal data is received, Cocoon first projects the per-query features, encoded by modality-specific encoders in the base detector, into a common representation space (Jing et al., 2021). Our uncertainty quantification is based on conformal prediction (Angelopoulos & Bates, 2021; Teng et al., 2023) that is widely adopted in practical applications due to its low computational complexity and interpretability (Luo et al., 2022a; Su et al., 2024a; Yang & Pavone, 2023). We propose a new conformal prediction method for heterogeneous representations by introducing the concept of a learnable surrogate ground truth, termed Feature Impression (FI), which enables feature-level uncertainty quantification. While traditional conformal prediction uses detection outputs and their ground truth labels in the output space—which are not available at intermediate stages of the model—our approach bypasses this limitation by employing FI. We demonstrate how FI can guide uncertainty quantification across heterogeneous representations, thus enabling fair comparisons. As such, Cocoon performs linear combination using uncertainty values, achieving object- and feature-level uncertainty-aware fusion.

The main contributions are as follows:

- We observed that ignoring object-level uncertainty in multi-modal object detection leads to suboptimal accuracy, particularly in long-tail conditions such as adverse weather or nighttime, highlighting the need for object-level uncertainty-aware fusion.
- Cocoon achieves object- and feature-level uncertainty-aware fusion by introducing uncertainty quantification for heterogeneous representations. The combination of a feature aligner and feature impression enables efficient and effective measurement of uncertainty at intermediate stages of the model. The resulting weights adjust each modality's contribution, allowing the model to prioritize the modalities with higher certainty to deliver more robust performance.
- In our evaluation on the nuScenes dataset (Caesar et al., 2019), Cocoon demonstrates notable improvement in both accuracy and robustness, consistently outperforming static and other adaptive fusion methods across normal and challenging scenarios, including natural and artificial corruptions. For example, in cases of camera malfunction, Cocoon improves the base model's mAP by 15% compared to static fusion.

Following the review of related work, we present a detailed analysis of the technical challenges in uncertainty-aware fusion within multi-modal perception systems. We then introduce the Cocoon framework and evaluate its effectiveness in addressing these challenges.

---

[1]While most existing work focuses on 2D perception or the fusion of RGB and depth camera data, we adapt these concepts to our situation.

## 2 RELATED WORKS

**Multi-Modal Perception.** Multi-modal solutions aim to enhance the accuracy and robustness of perception systems by leveraging diverse data fusion designs, which are typically categorized into early, intermediate, and late fusion. Early fusion combines features at the input level (Malawade et al., 2022b; Putzar et al., 2010; Huang et al., 2020; Sindagi et al., 2019; Vora et al., 2020), while late fusion integrates decisions from independent models at the object level (Sun et al., 2020; Huang et al., 2022; Xiang et al., 2023; Yeong et al., 2021). Favored in recent methods, intermediate fusion enhances accuracy by addressing discrepancies among the inductive biases of different modalities (Chen et al., 2017; Qin et al., 2023; Malawade et al., 2022a). These biases, which vary based on each modality's characteristics, are harmonized via joint learning, ensuring a more cohesive integration of multi-modal data. For example, cameras (Lu et al., 2021; Huang & Huang, 2022; Huang et al., 2021) rely on color and 2D shapes, while LiDAR (Shi et al., 2020; 2019) focuses on reflective properties and 3D shapes. Intermediate fusion combines these biases for a more robust environmental understanding. Techniques have evolved from LiDAR-centric fusion approaches, such as PointPainting and PointAugmenting (Vora et al., 2020; Wang et al., 2021), to more sophisticated architectures like the *Y-shaped* model, which fuses multi-modal features through concatenation, element-wise addition, or an adaptive fusion method (Liu et al., 2023b; Liang et al., 2022; Jiao et al., 2023). Recently, Transformer-based approaches (Chen et al., 2023; Prakash et al., 2021; Bai et al., 2022; Li et al., 2022b; Yan et al., 2023; Li et al., 2022a; 2023; Xie et al., 2023), inspired by DETR (Carion et al., 2020b; Zhu et al., 2020b; Wang et al., 2022), have become popular. They offer efficient detection paradigms using the Hungarian algorithm (Kuhn, 1955) (i.e., the predefined number of queries matched to objects), allowing flexible associations between modalities. Among these, FUTR3D (Chen et al., 2023) and TransFusion (Bai et al., 2022) stand out as the most foundational and representative transformer-based models, significantly influencing the current state-of-the-art in 3D object detection (Prakash et al., 2021; Li et al., 2022b;a; Yan et al., 2023; Li et al., 2023; Xie et al., 2023).

**Uncertainty Quantification.** Uncertainty quantification in machine learning enhances the trustworthiness and decision-making capabilities of systems by leveraging uncertainties present in both pre-trained models (epistemic uncertainty) and observed data (aleatoric uncertainty). Bayesian techniques are well-known methods in this domain, as they incorporate prior knowledge and update it with new data to estimate posterior distributions for model weights or input data. However, Bayesian methods are computationally intensive, making them less suitable for multi-sensory systems that require timely execution (Blundell et al., 2015; Kendall & Gal, 2017). In such contexts, alternative methods are preferred. One such method is direct modeling, primarily for data uncertainty, which modifies detection models by adding output layers to directly predict output variance (Feng et al., 2021; Su et al., 2024b; 2023). Another approach is Double-M Quantification, which estimates variability for both data and model uncertainty through resampling (Su et al., 2023). Conformal prediction provides prediction intervals using a nonconformity measure and a calibration dataset for addressing both data and model uncertainty (Angelopoulos & Bates, 2021; Teng et al., 2023; Javanmardi et al., 2024). Conformal prediction, with its low computational complexity, distribution-free nature, provability, and interpretability, is most optimal for practical use (Luo et al., 2022a; Su et al., 2024a; Yang & Pavone, 2023).

## 3 CHALLENGES IN UNCERTAINTY-AWARE FEATURE FUSION

**Object-Level Uncertainty Variation Phenomenon.** Assessing each modality's uncertainty, defined by how reliable its information is about the input scene, is challenging due to the variety of objects within the scene. Additionally, uncertainties are influenced by various factors, including data informativeness and unusualness relative to a pre-trained perception model, making the assessment non-trivial.

We conduct an experiment to empirically demonstrate the impact of diverse uncertainties, as shown in Fig. 2. We apply a baseline perception model (FUTR3D (Chen et al., 2023)), trained to fuse camera and LiDAR data in a 1:1 ratio, to various traffic scenes. We tune the fusion ratio and observe its impact on model effectiveness, measured by the average confidence score. Interestingly, the 1:1 fusion ratio between 2D and 3D data proved to be suboptimal in most scenarios. In daytime scenarios, camera data on distant and small objects is more informative than LiDAR data, suggesting an optimal fusion ratio of 7:3. Conversely, for nearby objects, greater accuracy is achieved when the contribution of LiDAR is increased relative to the camera. Moreover, LiDAR sensors are more effective than cameras at nighttime, indicating that more weight should be assigned to LiDAR inputs to improve perception accuracy.

Overall, this imbalance in the contribution of 2D and 3D data is common and related to various factors such as lighting conditions, object sizes, and distances to the ego agent. These observations

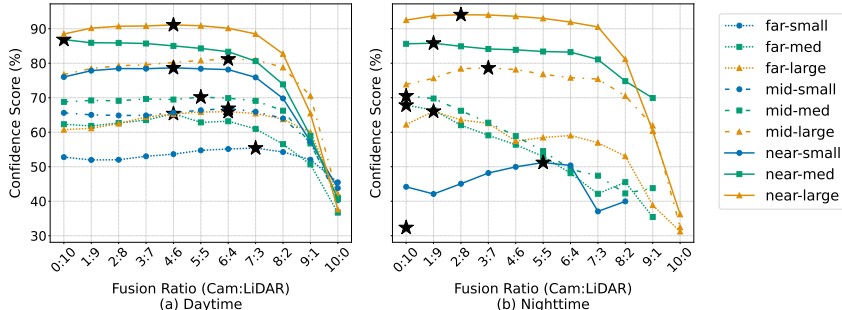

Figure 2: Impact of fusion ratio on average confidence scores under various lighting conditions and object configurations. The black stars denote the optimal camera-to-LiDAR fusion ratios achieving the highest scores for each configuration. Object configurations are categorized based on two attributes: object size and distance to the ego agent. Object sizes are classified into three categories: small ($< 2m$), medium ($2$-$4m$), and large ($> 4m$). Similarly, distances are segmented into three ranges: near ($< 20m$), mid-distance ($20$-$40m$), and far ($> 40m$).

highlight the dynamic nature of object-level uncertainties and the need for specific dynamic weighting for each object based on its unique characteristics. To address this, Cocoon quantifies uncertainties for per-object features and dynamically adjusts the weights accordingly.

**Undefined Consistent Basis for Comparison.** Although existing methods like Cal-DETR (Munir et al., 2023), which computes variance along transformer decoder layers, can quantify uncertainties within a single modality, projecting uncertainties from multi-modal data into a consistent and comparable space remains challenging. This step is crucial for effective multi-modal, uncertainty-aware fusion, where the alignment and comparability of data from different modalities are essential.

The challenge arises from the distinct representation spaces of multi-modal data: 2D data provides color and texture information, whereas 3D data offers depth and geometric insights. We address this challenge through Feature Impression-based feature alignment, motivated by Jing et al. (2021).

## 4 COCOON

In this section, we introduce Cocoon, an object- and feature-level modality fusion solution. Cocoon employs a query-based architecture with a fixed number of predefined queries, each representing a potential object, enabling it to extract object-level features without relying on final detection outputs (Carion et al., 2020a; Chen et al., 2023; Bai et al., 2022). Specifically, we use FUTR3D (Chen et al., 2023) and TransFusion (Bai et al., 2022)—two foundational and representative query-based models—as our base models. Built on these architectures, Cocoon operates in two phases: an online phase that performs adaptive fusion on test data, and an offline phase that prepares for uncertainty quantification (i.e., conformal prediction) through training and calibration stages.

Fig. 3 illustrates the online mechanism, where we first align per-object features into a common representation space (Sec. 4.2) and then quantify uncertainties for each pair of features (Sec. 4.3). These uncertainties are used as weights ($\alpha$ and $\beta$) in the adaptive fusion. For the offline mechanism, we first split the training set into a proper training set and a calibration set. The proper training set is used to train both the base model components and the parameters for uncertainty quantification. Using the calibration set, we create a comparison group containing task-related scores for each calibration sample, which are then used to assess how unusual the online input data is, thereby facilitating uncertainty quantification (Sec. 4.4).

We begin by revisiting the inspiring work, Conformal Prediction (Angelopoulos & Bates, 2021) and Feature CP (Teng et al., 2023), and their limitations in a multi-modal setting. We then explain how we can fairly quantify the uncertainties of multi-modal features on a consistent basis.

### 4.1 PRELIMINARIES

**Conformal Prediction (CP).** Conformal prediction, referred to as Basic CP, is an algorithm that outputs a prediction interval, statistically guaranteeing the inclusion of the ground truth (GT) for a given test input (Angelopoulos & Bates, 2021; Teng et al., 2023), rather than merely providing a single prediction value. This prediction interval is established with respect to a significance level ($\Lambda$), ensuring that the interval contains the GT with a probability of at least 1-$\Lambda$, as follows:

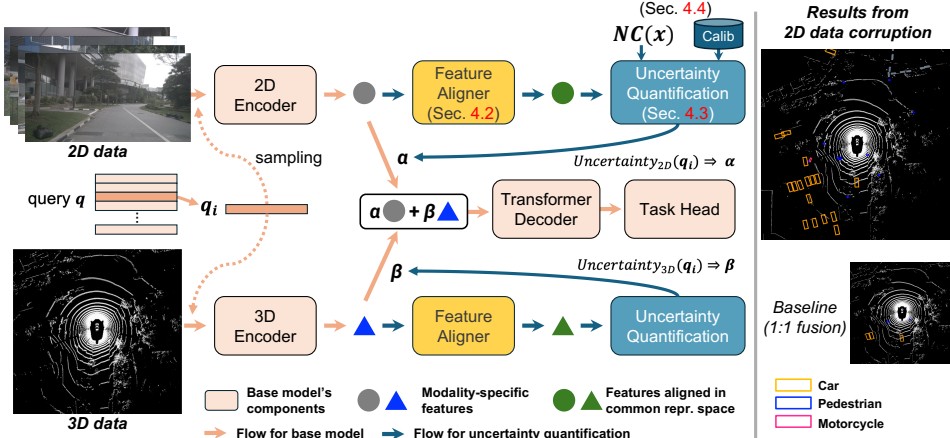

Figure 3: Cocoon Online Procedure (left) and Example Results (right): Cocoon operates on top of base model components. The feature aligner projects per-object features (●, ▲) into a common representation space. Next, uncertainties are estimated for each feature pair (●, ▲) and converted into weights ($\alpha$ and $\beta$) for adaptive fusion. These weights amplify/attenuate the contribution of each modality's original feature (●, ▲) to the fused feature, which is then used in the basemodel's decoder.

Table 2: Summary of Main Symbols: Meaning and Purpose

| Symbol | Meaning | Purpose | Section |
|---|---|---|---|
| $\alpha, \beta$ | Key symbols; $\alpha$ represents the weight of one modality (e.g., camera), and $\beta$ represents the weight of another (e.g., LiDAR). | Uncertainty-based weights (Dynamic) | §4.3, §5.2, Fig. 3, Fig. 6, Fig. 7 |
| $\delta, \zeta, \eta$ | Coefficients in the training loss function. | Hyperparameter | §5.1, Eq. 3 |
| $\Lambda$ | Significance value used to explain the concept of conformal prediction. | Analysis | §4.1, §5.3, Table 4 |

$$P(y' \in \mathcal{I}_{1-\Lambda}(x')) \geq 1 - \Lambda,$$

where $\mathcal{I}_{1-\Lambda}(x')$ represents the prediction interval for online test input $x'$.

The process begins with preparing a training set, a calibration set, a pre-trained model, and the establishment of a nonconformity function. During the offline stage, the nonconformity score $NC_i = \text{Distance}(p_i, q_i)$ is computed for each calibration instance, where $p_i$ and $q_i$ denote the predicted output and the GT of instance $i$, respectively. The pool of scores is used to gauge how unusual online test data is. In the online stage, the algorithm uses $1 - \Lambda$ to compute the tightest prediction interval that includes the GT with at least $1 - \Lambda$ confidence. This is achieved by calculating the $(1 - \Lambda)$-th quantile from the distribution derived from $NC_i$, for $i \in$ calibration set. Unlike Basic CP, which quantifies output-space uncertainty using GT labels on calibration samples, we address feature-space uncertainty without relying on modality-specific GT labels.

**Feature Conformal Prediction (Feature CP).** Teng et al. (2023) extends conformal prediction to the feature level using a pre-trained model $M$, structured into $f$ (encoder) and $g$ (decoder; task head), formulated as $Y = M(X) = (g \circ f)(X)$, with $\circ$ denoting the composition operator. This method measures nonconformity scores directly at the feature level via the encoder $f$. For nonconformity measure, a surrogate ground truth $v^*$ is identified through an iterative search (see Fig. 4 (a)) to serve as a proxy for the normal GT within a norm-based nonconformity function: $NC_i = \|f(x_i) - v^*_{x_i}\|$. Here, $v^*$ is meticulously calibrated to closely resemble $g^{-1}(y_i)$. However, applying feature conformal prediction directly to our problem poses challenges, primarily because it assumes *homogeneous* representation across target features—an assumption that does not hold in multi-modal settings.

## 4.2 FEATURE IMPRESSION-BASED FEATURE ALIGNER

**Core Issue: Lack of Modality-Dedicated Decoder.** As illustrated in Fig. 4, a dedicated decoder $g$ often does not exist in multi-modal detection models. The absence of $g$ highlights a core issue, as $g$ plays a crucial role in facilitating the bidirectional movement between output space and feature space. To address this issue, *we perform all operations directly at the feature level*, eliminating the conventional move in/out mechanism. Our intuition is that effectively aggregating multi-modal

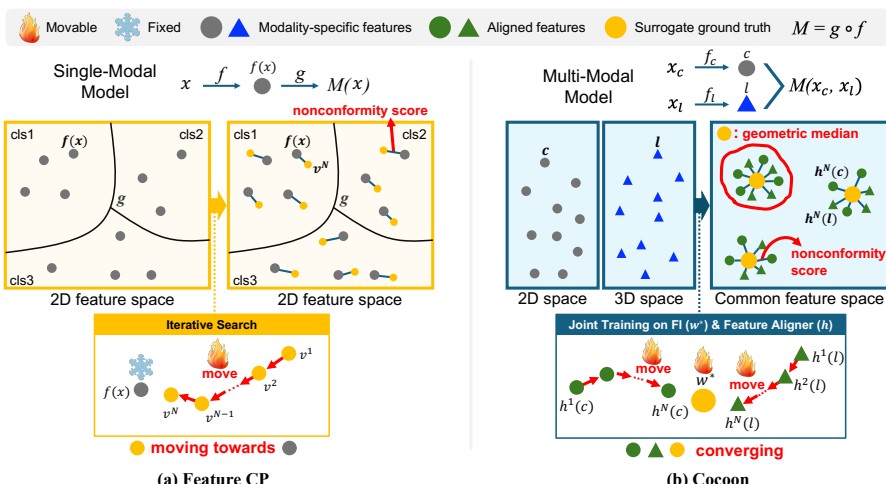

Figure 4: Feature CP **vs.** Cocoon. In the **offline** stage with calibration data, Feature CP identifies the surrogate ground truth (🟡) for each feature (⚫) through iterative search. Each 🟡 is derived using the real ground truth label in the output space and the decoder $g$ (serving as a classifier). However, in a multi-modal setting, each feature lacks a modality-specific $g$. To resolve this, Cocoon leverages joint training of the feature aligner (which projects heterogeneous features (⚫, ▲) into a common representation space) and the surrogate ground truth (🟡 – termed FI). Our proposed training objective (Eq. 3) ensures that each FI becomes the *geometric median* of aggregated features for valid uncertainty quantification. In the **offline** stage, the calibration samples' nonconformity scores (i.e., distance) are collected and used as a comparison target to gauge the uncertainty of online test inputs. In the **online** stage with test data, while Feature CP performs iterative search for 🟡, Cocoon saves time by directly projecting input features through our feature aligner $h$ and using a pre-trained 🟡.

features into a high-dimensional sphere substantially reduces the need for $g$. The following sections demonstrate how conformal prediction can operate efficiently with multi-modal features without $g$.

Based on our intuition, we initiate the process by aligning multi-modal features within a unified representation space using a simple multi-layer perceptron (MLP), inspired by Jing et al. (2021). The key aspect of our approach is the learnable surrogate GT and the joint training of both the surrogate GT and the feature aligner. For clarity, we introduce the new term, feature impressions (FIs; 🟡 in Fig. 4 (b)), with each FI representing a class or a set of similar features and serving as a surrogate GT.

**Conditions for FI.** As shown in Fig. 4, Feature CP identifies the surrogate GT $v^*_{x_i}$ as the **nearest** feature (🟡 in Fig. 4 (a)) to the input sample $x_i$ (⚫), obtained through iterative search. In the feature space, each $x_i$ has its unique $v^*_{x_i}$, which eventually corresponds to $y_i$ in the output space. In other words, although they measure nonconformity scores in feature space, their final uncertainty values (w.r.t. $y_i$) are quantified in the output space. Note that if this is a classification task, $y_i$ (e.g., class label) is **shared** by multiple $x_i$s belonging to the same class. Based on our intuition, we aim to remove bidirectional movement between feature and output spaces and perform uncertainty quantification directly in the feature space. To achieve this, our surrogate ground truth should satisfy two conditions: (1) we have to identify a **common** ground truth for the aligned multi-modal features; (2) given evidence in Feature CP, each FI node (🟡) should be **as close as possible** to the aligned feature (⚫, ▲) to serve as an effective surrogate GT.

Assuming all distances between an FI node and its corresponding aligned features are **distinct positive** values, the FI node should be the **unique minimizer** of the sum of all the distances to serve as the effective surrogate GT. This strategic placement establishes the each FI as the **geometric median**, effectively capturing the essence of the associated features' distribution within the unified representation space. Consequently, we can define our nonconformity function as:

$$NC_i = \|(h \circ f)(x_i) - w^*_{x_i}\|, \tag{1}$$

where $h$ represents the feature aligner, and $w^*_{x_i}$ denotes the $x_i$'s corresponding FI node. This positioning not only enhances efficiency by eliminating the costs associated with $g$ but also simplifies the overall uncertainty quantification process.

### 4.3 UNCERTAINTY QUANTIFICATION WITH MULTI-MODAL FEATURE SET

Uncertainty quantification entails identifying the most probable FI node at each transformer decoder layer, leveraging insights from the nonconformity function in Eq.1 (see Appendix J for further details).

---

***Finding 1:*** Feature uncertainty significantly affects the NC score of the top-1 FI node.

***Finding 2:*** Higher feature uncertainty in transformers with multiple decoder layers leads to significant instability in the top-1 FI node across layers.

---

Based on these observations, our primary uncertainty value is defined using the metric (p-value) employed in conformal prediction (Vovk et al., 2005):

$$\mathcal{P}_{\text{mod}} = \frac{|\{x \in NC_{\text{calib}} : x \geq nc_{\text{mod}}\}|}{|NC_{\text{calib}}|}, \tag{2}$$

where $NC_{\text{calib}}$ represents the pool of nonconformity scores computed from all aligned calibration samples, and $nc_{\text{mod}}$ denotes the nonconformity score of the current online input data's feature. Leveraging this, the final weight for each modality is calculated as:

$$\mathcal{W}_{\text{mod}} = \frac{\mathcal{Q}_{\text{mod}}\mathcal{S}_{\text{mod}}}{\mathcal{Q}_{\text{mod}}\mathcal{S}_{\text{mod}} + \mathcal{Q}_{\text{other}}\mathcal{S}_{\text{other}}},$$

where $\mathcal{Q}_{\text{mod}} = \frac{\mathcal{P}_{\text{mod}}}{\mathcal{P}_{\text{mod}}+\mathcal{P}_{\text{other}}}$, and $\mathcal{S}_{\text{mod}}$ denotes the stability score, measured by the frequency of top-1 FI changes across decoder layers. *other* refers to the modality opposite to *mod*. The resulting uncertainties are used as weights ($\alpha$ and $\beta$ in Fig. 3) in adaptive fusion.

### 4.4 TRAINING AND CALIBRATION STAGES

The training process consists of two stages: the first stage involves training the baseline model using a proper training set, ensuring the pre-trained model has no exposure to the calibration samples. The second stage involves joint training of the feature aligner and FIs, while keeping the base model components frozen. Since the first stage follows the same procedure as the original base model's training (Chen et al., 2023; Bai et al., 2022), we focus on the second stage in this section.

Our nonconformity function (Eq. 1) requires each FI node to act as the *geometric median* for its associated features. A specifically tailored regularizer is crucial to maintain this positioning.

**Joint Training of FI and Feature Aligner.** The ***Weiszfeld's algorithm*** stands out as a notable solution method for the geometric median problem (Chandrasekaran & Tamir, 1989). For a point $y$ to qualify as the geometric median, it must meet the following criterion:

$$0 = \sum_{i=1}^{m} \frac{x_i - y}{\|x_i - y\|}$$

We can apply this criterion to our problem setting where we have a set of $2N$ features:

$$h_{ci} := h \circ f_c(x_{ci}), i = 1, \ldots, N, \quad h_{li} := h \circ f_l(x_{li}), i = 1, \ldots, N$$

where $f_c$ and $f_l$ are encoders in the model, and $x_{ci}$ and $x_{li}$ represent input data from multiple sensors. With these terms defined, our training objective consists of three terms: $\mathcal{L}_{center}$ for aggregating different modality features around their corresponding FIs ($w^*$), $\mathcal{L}_{geomed}$ for ensuring that the FIs serve as the geometric median of the aligned features, and $\mathcal{L}_{separate}$ for increasing the distance between the positions of $w^*$, ensuring clear separation between FIs.

$$\mathcal{L} = \delta \underbrace{\sum_{i=1}^{N}(\|w^* - h_{ci}\| + \|w^* - h_{li}\|)}_{\mathcal{L}_{center}} + \zeta \underbrace{\left\| \sum_{i=1}^{N}\left(\frac{h_{ci} - w^*}{\|h_{ci} - w^*\|} + \frac{h_{li} - w^*}{\|h_{li} - w^*\|}\right) \right\|^2}_{\mathcal{L}_{geomed}} - \eta \underbrace{\sum_{j=1}^{C}\sum_{k=j+1}^{C}\|w_j^* - w_k^*\|^2}_{\mathcal{L}_{separate}} \tag{3}$$

Here, $C$ denotes the number of FI nodes (e.g., the number of classes). This loss function is used to jointly train feature aligner ($h$) and FIs ($w^*$). See Appendix C for the derivation of Eq. 3.

**Calibration.** This process involves computing nonconformity scores (Eq. 1) for each calibration instance using the pre-trained FIs and feature aligner $h$. The pool of calibration scores is then used to assess how unusual the features of the online input data are, facilitating uncertainty quantification. Further details about the offline process are provided in Appendix H.

Table 3: Accuracy (mAP) Comparison with Baseline Methods Under Different Scenarios. Orange represents original nuScenes validation set, green indicates natural corruptions collected from real-world driving environments, and blue corresponds to artificial corruptions presented in existing work.

| Methods | Type | No Corruption | Rainy Day | Clear Night | Rainy Night |
|---|---|---|---|---|---|
| Base$_{FUTR3D}$ (Chen et al., 2023) | Static | 66.16 | 68.34 | 44.51 | 27.26 |
| AdapNet* (Valada et al., 2017) | MoE-based | 62.33 | 62.82 | 41.74 | 21.58 |
| Zhu et al. (2020a) | Self-attention | 63.72 | 64.22 | 41.17 | 20.79 |
| Zhang et al. (2023) | Late Fusion | 63.93 | 64.64 | 42.92 | 22.56 |
| Cal-DETR (T) (Munir et al., 2023) | Uncertainty-aware | 66.24 | 68.37 | 44.38 | 26.29 |
| Cal-DETR (Q) (Munir et al., 2023) | Uncertainty-aware | 66.38 | 68.42 | 44.61 | 27.30 |
| Cocoon (Ours) | Uncertainty-aware | **66.80** | **68.89** | **45.68** | **27.98** |

| Methods | Type | Point Sampling (L) | Random Noise (C) | Blackout (C) | Sensor Misalign. |
|---|---|---|---|---|---|
| Base$_{FUTR3D}$ (Chen et al., 2023) | Static | 65.17 | 60.39 | 45.13 | 53.16 |
| AdapNet* (Valada et al., 2017) | MoE-based | 61.40 | 55.83 | 36.43 | 46.04 |
| Zhu et al. (2020a) | Self-attention | 62.60 | 56.07 | 37.82 | 45.91 |
| Zhang et al. (2023) | Late Fusion | 63.24 | 56.32 | 39.39 | 49.61 |
| Cal-DETR (T) (Munir et al., 2023) | Uncertainty-aware | 65.29 | 60.40 | 44.39 | 54.87 |
| Cal-DETR (Q) (Munir et al., 2023) | Uncertainty-aware | 65.40 | 60.56 | 45.25 | 54.54 |
| Cocoon (Ours) | Uncertainty-aware | **65.89** | **61.83** | **51.87** | **55.61** |

Table 4: Accuracy Breakdown w.r.t. Distance.

| Corruption | Near (< 20m) Static | Near (< 20m) Ours | Mid (20-40m) Static | Mid (20-40m) Ours | Far (> 40m) Static | Far (> 40m) Ours |
|---|---|---|---|---|---|---|
| No Corruption | 72.3 | **72.6** | 53.9 | **54.3** | 7.4 | **7.7** |
| Point Sampling (L) | 72.2 | **72.5** | 53.5 | **53.7** | 7.0 | **7.4** |
| Random Noise (C) | 69.8 | **70.9** | 49.4 | **50.4** | 7.4 | **7.9** |
| Blackout (C) | 67.0 | **69.4** | 37.5 | **42.8** | 3.3 | **4.5** |
| Misalignment | 68.4 | **69.0** | 46.3 | **47.1** | 4.1 | **4.4** |

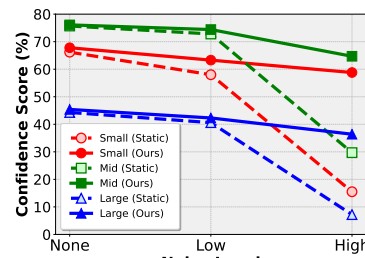

Figure 5: Confidence Score Change.

## 5 EVALUATION

In this section, we evaluate Cocoon's accuracy and robustness across diverse scenarios.

### 5.1 EVALUATION SETUP

**Dataset.** Our evaluations utilize the nuScenes dataset (Caesar et al., 2019), a comprehensive autonomous driving dataset collected from vehicles equipped with a 32-beam LiDAR and 6 RGB cameras. For conformal prediction, we partition the original training dataset into a proper training set and a calibration set with a 6:1 ratio. Details on our dataset are discussed in Appendix A.

**Base Models.** Cocoon framework is based on 3D object detectors, FUTR3D (Chen et al., 2023) and TransFusion (Bai et al., 2022), known for their unique architectures and influence on recent advancements in the field (Xie et al., 2023; Yan et al., 2023). This section focuses on FUTR3D, with results for TransFusion detailed in Appendix F. FUTR3D encodes 2D and 3D data using ResNet101 He et al. (2016) and VoxelNet Zhou & Tuzel (2018), respectively, fuses the encoded features via concatenation, and then outputs prediction results after passing through six transformer decoder layers. To ensure compatibility with Cocoon, we replace the concatenation with an element-wise sum following the integration of the feature aligner and FI nodes.

**Corruption Scenarios.** We simulate natural and artificial corruptions that impact online inference results. Natural corruptions include adverse weather and lighting variations, categorized from the nuScenes validation set. For artificial corruptions, 'blackout' simulate camera malfunctions, while other artificial corruptions, identified as challenging conditions (Yu et al., 2023; Dong et al., 2023), represent realistic disturbances. Detailed corruption configurations are provided in Appendix B.

**Baselines.** AdapNet* (Valada et al., 2017) uses an MoE approach with a trainable gating network; Zhu et al. (2020a) applies self-attention to each modality feature to determine weights; Zhang et al. (2023) employs late fusion with an enhanced Non-Maximum Suppression for output-level adaptive fusion; Cal-DETR (Munir et al., 2023) measures uncertainty using variance in transformer decoder layers. Cal-DETR (T) determines the weights at the tensor level, while Cal-DETR (Q) determines them at the query level, allowing evaluation of object-level adaptive fusion. See Appendix I for details.

**Training and Inference Details.** For computational resources, 4 A40 GPUs were used for training, and 1 RTX 2080 for calibration and inference. The training process consists of two stages: the first stage involves training the baseline model from scratch using a proper training set, ensuring the pre-trained model has no knowledge of the calibration samples. The second stage involves training the feature aligner and feature impression nodes. We employ an 8-layer MLP for feature

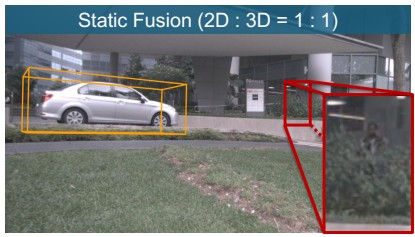 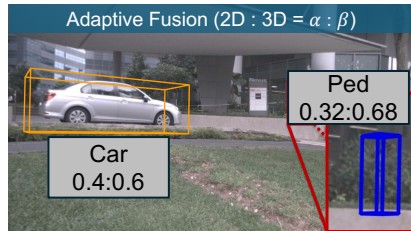

(a) Occluded pedestrian ('Ped.') detected by adaptive fusion.

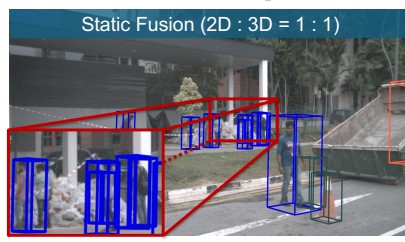 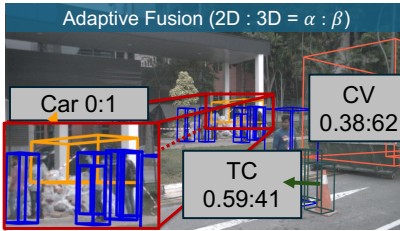

(b) Improved detection on the occluded car (left) and the construction vehicle ('CV'; right). 'TC': traffic cone.

Figure 6: Qualitative Comparison between Static Fusion and Cocoon on Challenging Objects.

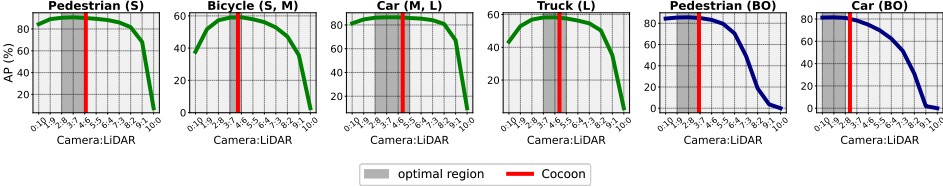

Figure 7: Alignment of Cocoon's resulting weights ($\alpha$:$\beta$, red lines) with optimal regions (grey area). The optimal region is determined as the area within a margin of 0.5 from the peak AP value. "S": small, "M": medium, "L": large, "BO": blackout.

aligner to align modality-specific features to a 128-dimensional representation space. Each FI node indicates per-class feature space within this common representation space. For the first training stage, the baseline models were trained using the same hyperparameters (e.g., learning rate) as those used in original models. In the second stage, a single batch size was used without changing other hyperparameters. Inference was conducted with a single batch size. In Eq. 3, we set $\delta = \frac{5}{\text{num\_queries}}$, $\zeta = \frac{3}{\text{num\_queries}}$, and $\eta = \frac{1}{7 \times \text{num\_queries}}$. See Appendix D for these coefficient values and further details.

## 5.2 EVALUATION ON FUTR3D

**Quantitative Comparison.** Table 3 presents Cocoon's accuracy and robustness across various scenarios. Cocoon consistently outperforms other baselines in both normal and unpredictable environments. For further analysis, we compare the static 1:1 fusion (the original fusion method in the base model) and Cocoon from different perspectives. In Table 4, we break down accuracy based on object locations (distance from the ego agent). Cocoon demonstrates better accuracy in all cases. Notably, the improvements are more significant for mid-distance and far-away objects, which tend to have smaller pixels and sparser points, emphasizing Cocoon's robustness for more challenging objects.

We also evaluate Cocoon's robustness to varying noise levels to assess its generalizability. Fig. 5 presents results under different severities of random noise interference applied to 2D data. "None" represents normal conditions, "Low" introduces uniform noise in the range of -50 to 50, and "High" adds noise in the range of -150 to 150. For fair comparison, we only evaluate commonly detected objects across both static fusion and Cocoon fusion at all noise levels. Even under extreme noise, Cocoon exhibits less perturbation in the confidence scores of objects, demonstrating its ability to mitigate the impact of corrupted modalities and maintain detection accuracy in diverse conditions.

**Alignment between Cocoon and Optimal Region.** Our primary hypothesis is that optimally weighting different modalities can enhance detection performance, leading to improved APs. To validate this, we assess how well the $\alpha$ and $\beta$ determined by Cocoon align with the optimal weights. Fig. 7 illustrates this alignment across various object categories in both normal and blackout scenarios. The alignment is consistently observed across diverse object sizes in both conditions, supporting our hypothesis.

Table 5: Comparison of NC Function Effectiveness Across 10 Datasets ($\Lambda = 0.1$). Run 5 times for mean/standard deviation. 'Abs Diff' refers to the absolute difference between the mean and the ideal target of 90%, indicating NC score validity (lower is better). Our method consistently outperforms others, demonstrating the effectiveness of FI, a geometric median-based surrogate ground truth.

| Dataset | BasicCP (Angelopoulos & Bates, 2021) | Abs Diff ↓ | Feature CP (Teng et al., 2023) | Abs Diff ↓ | Ours | Abs Diff ↓ |
|---|---|---|---|---|---|---|
| Meps19 | $90.51 \pm 0.25$ | 0.51 | $90.55 \pm 0.21$ | 0.55 | $90.24 \pm 0.65$ | **0.24** |
| Meps20 | $89.80 \pm 0.75$ | 0.20 | $89.76 \pm 0.74$ | 0.24 | $90.02 \pm 0.72$ | **0.02** |
| Meps21 | $89.92 \pm 0.66$ | 0.08 | $89.94 \pm 0.66$ | 0.06 | $90.05 \pm 0.65$ | **0.05** |
| Community | $89.82 \pm 1.15$ | 0.18 | $89.62 \pm 1.01$ | 0.38 | $90.18 \pm 1.30$ | **0.18** |
| Facebook1 | $90.12 \pm 0.40$ | 0.12 | $90.08 \pm 0.40$ | 0.08 | $89.98 \pm 0.22$ | **0.02** |
| Facebook2 | $89.95 \pm 0.16$ | 0.05 | $89.96 \pm 0.18$ | 0.04 | $89.98 \pm 0.23$ | **0.02** |
| Star | $90.30 \pm 1.42$ | 0.30 | $90.35 \pm 1.21$ | 0.35 | $90.02 \pm 1.73$ | **0.02** |
| Blog | $90.11 \pm 0.3$ | 0.11 | $90.06 \pm 0.39$ | **0.06** | $89.94 \pm 0.31$ | **0.06** |
| Bio | $90.23 \pm 0.3$ | 0.23 | $90.23 \pm 0.37$ | 0.23 | $90.13 \pm 0.21$ | **0.13** |
| Bike | $89.66 \pm 1.04$ | 0.34 | $89.62 \pm 0.81$ | 0.38 | $90.25 \pm 1.09$ | **0.25** |

**Qualitative Results.** As shown in Fig. 6, Cocoon dynamically adjusts the weights between 2D and 3D data, thereby improving detection performance under complex conditions (e.g., occluded objects).

**Limitations.** The primary limitation is the runtime overhead, as discussed in Appendix K. When using FUTR3D, we observe a 0.5-second increase in latency. To mitigate this, computing nonconformity scores only in the last three decoder layers reduces latency while maintaining better robustness than static fusion. In the section, we also present further methods to minimize this latency gap (e.g., reducing the number of queries). Another limitation is the presence of meaningless queries. Transformer-based architectures, like FUTR3D, often predefine more queries (900) than necessary (up to 200). We found that when $\alpha$ exceeds 0.7, which leads to suboptimal results in all cases in Fig. 2, the meaningless queries predominate, impacting final output due to layer normalization. To address this issue, a clipping strategy is applied, whereby meaningless queries undergo static fusion to ensure stable performance. Fig. 6 shows that queries matched to actual objects remain unaffected by clipping.

## 5.3 VALIDITY OF THE PROPOSED NONCONFORMITY FUNCTION

In this section, we aim to quantify uncertainty at the feature level, similar to Feature CP (Teng et al., 2023), but with a different approach. The effectiveness of the nonconformity (NC) function is quantitatively validated through a coverage check, a method widely used in conformal prediction research. Our evaluation examines the alignment between the prediction interval at a given significance level ($\Lambda$) and the actual ground truth coverage. For this regression task, our feature aligner projects all features so that they cluster around a single FI node.

Following previous studies, we also use 10 uni-dimensional real-world datasets for validation. Several datsets (bio, bike, community, facebook1, and facebook2) are sourced from the UCI Machine Learning Repository (Kelly et al., 2023; Singh, 2016). Additionally, we utilize data from the Blog Feedback dataset (Buza, 2014), the STAR dataset (Achilles et al., 2008), and the Medical Expenditure Panel Survey datasets (meps19, meps20, and meps21) (Cohen et al., 2009).

In Table 5, our nonconformity function closely aligns with the specified significance level (0.1), consistently including the ground truth in the prediction interval 90% of the time. This demonstrates the effectiveness of our method, leveraging FI (geometric median-based surrogate ground truth), with empirical coverage highlighting the model's proper calibration and effective uncertainty quantification. Qualitative validation on 3D object detection is provided in Appendix **??**.

## 6 CONCLUSION & FUTURE WORK

In this work, we introduced Cocoon, an uncertainty-aware adaptive fusion framework designed to address the limitations of existing modality fusion methods in 3D object detection. By achieving object- and feature-level uncertainty quantification via conformal prediction, Cocoon enables more effective fusion of heterogeneous representations, enhancing accuracy and robustness across various scenarios.

For future work, we plan to extend Cocoon to accommodate additional types and numbers of modalities, such as fusing camera, LiDAR, and radar for autonomous vehicles and robotics, or combining vision and language for VLMs (Yang et al., 2023; Liu et al., 2024; Team et al., 2023). This broader applicability will enable systems to integrate diverse modalities more effectively, enhancing their ability to interpret complex, challenging environments in ways that mirror human cognition.

ACKNOWLEDGEMENTS

We express our gratitude to our area chairs and anonymous reviewers for their invaluable feedback. This work was supported by NSF under the National AI Institute for Edge Computing Leveraging Next Generation Wireless Networks, Grant 2112562, in addition to grants CMMI-2038215 and CNS-2321532.

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

## A    DATASET DETAILS

Our evaluations utilize the nuScenes dataset, a comprehensive autonomous driving dataset collected from vehicles equipped with a 32-beam LiDAR and 6 RGB cameras (Caesar et al., 2019). This dataset includes 700 training scenes and 150 validation scenes, each lasting 20 seconds and annotated at a frequency of 2 Hz.

Table 6: Scene Distribution

| Dataset | Day | Night | Rate (D/N) |
|---|---|---|---|
| **Original** | 616 | 84 | 7.3 |
| **Training** | 528 | 72 | 7.3 |
| **Calibration** | 88 | 12 | 7.3 |

In preparation for conformal prediction, we have partitioned the training data into a proper training set and a calibration set, using a 6:1 ratio, as shown in Table 6. The proper training set is used to train the parameters of the base detection model, feature aligner, and FI, while the calibration set is applied during the calibration phase.

One crucial assumption of conformal prediction is *exchangeability*, which states that the distribution of a sequence of instances remains invariant under any permutation of these instances. Since different scenes are distinct enough to be considered exchangeable (Luo et al., 2022b), we generate a dataset of exchangeable calibration samples by sampling a single instance (i.e., frame) at random from each scene belonging to calibration set.

## B    CONFIGURATIONS FOR ARTIFICIAL CORRUPTIONS

In this paper, we investigate various artificial corruption scenarios, referencing recent benchmarking studies (Yu et al., 2023; Dong et al., 2023). We explore two types of 2D data corruptions (applied to six images), one type of 3D data corruption, and sensor misalignment scenarios.

For 2D data corruption, camera blackout conditions simulate potential camera malfunctions by converting images into entirely black frames. Additionally, we introduce random noise interference to the six images by adding random noise in the range of -50 to 50 to the original RGB values, where each pixel's value ranges from 0 to 255.

For 3D data corruption, we randomly delete 30% of the points in one frame of LiDAR data. LiDAR density decrease is a common corruption in LiDAR data, shown in various datasets (Dong et al., 2023; Sun et al., 2022; Kong et al., 2023). A 30% density decrease, as shown in Dong et al. (2023), significantly impacts accuracy (dropping by 0.39 to 0.5) in 2D-3D fusion models. This corruption arises from adverse weather, surface reflectivity, and sensor resolution issues Dong et al. (2023); Kong et al. (2023), making it representative of long-tail inputs in public datasets (Geiger et al., 2012; Caesar et al., 2019). Furthermore, due to the large memory usage of LiDAR point clouds, many works use random sampling at higher rates (Li et al., 2021; Yang et al., 2024; Cho et al., 2023), making density decrease a more frequent use case.

Sensor misalignment occurs due to calibration errors, malfunctioning components, or aging sensors, causing spatial or temporal misalignment between sensor data (Dong et al., 2023; Yu et al., 2023). Synchronization issues can arise from external/internal factors, such as spatial misalignment caused by severe vibrations when driving on bumpy roads or temporal misalignment due to clock synchronization module malfunctions. Yu et al., 2023 provides an in-depth investigation into both spatial and temporal alignments. Referencing these works, we add random Gaussian noise to the calibration matrices. In the case of the nuScenes dataset, the calibration matrix, *lidar2cam*, is not normalized, showing a range of (-1463.53, 1516.51) with a mean of -127.18 and a standard deviation of 516.67. Like the point density drop, we introduce 33-34% noise, (-500, 500), which leads to a meaningful accuracy drop.

## C    DERIVATION OF TRAINING OBJECTIVE

The challenge of identifying a point that minimizes the sum of unsquared distances to all other points characterizes the geometric median problem. The ***Weiszfeld's algorithm*** stands out as a notable solution method for this problem (Chandrasekaran & Tamir, 1989). For a point $y$ to qualify as the geometric median, it must meet the following criterion:

$$0 = \sum_{i=1}^{m} \frac{x_i - y}{\|x_i - y\|}$$

This formula confirms that $y$, distinct from all other points, accurately represents the geometric median, underpinning the efficacy of our nonconformity function.

Based on this algorithm, we can apply this to our problem setting where we have a set of $2N$ features:

$$h_{ci} := h \circ f_c(x_{ci}), i = 1, \ldots, N, \quad h_{li} := h \circ f_l(x_{li}), i = 1, \ldots, N$$

where $f_c$ and $f_l$ are encoders in the model, and $x_{ci}$ and $x_{li}$ represent input data from multiple sensors. With these terms defined, the goal is to ensure that $w^*$ is the *geometric median* of this set of features:

$$w^* = \arg \min_w \sum_{i=1}^{N} \left( \|w - h_{ci}\| + \|w - h_{li}\| \right). \tag{4}$$

Assuming $w^*$ is distinct from each feature $w^* \neq h_{ci}, w^* \neq h_{li}$, then the sufficient and necessary optimality condition for problem equation 4 is

$$0 = \sum_{i=1}^{N} \left( \frac{h_{ci} - w^*}{\|h_{ci} - w^*\|} + \frac{h_{li} - w^*}{\|h_{li} - w^*\|} \right). \tag{5}$$

Therefore, we can design a loss function to penalize the violation of equation 5

$$\mathcal{L}_{geomed} = \left\| \sum_{i=1}^{N} \left( \frac{h_{ci} - w^*}{\|h_{ci} - w^*\|} + \frac{h_{li} - w^*}{\|h_{li} - w^*\|} \right) \right\|^2. \tag{6}$$

If $\mathcal{L}_{geomed}$ is zero, it confirms that $w^*$ satisfies the geometric median condition as defined in equation 4. As training progresses, the positions of $w^*$ tend to converge, which could potentially cause ambiguities in distinguishing between them. To address this issue, we introduce a penalty, $\mathcal{L}_{separate}$, for increasing distances between the positions of $w^*$, aiming to maintain clear separations. Consequently, our total loss function incorporates this additional penalty:

$$\mathcal{L} = \delta \underbrace{\sum_{i=1}^{N}(\|w^* - h_{ci}\| + \|w^* - h_{li}\|)}_{\mathcal{L}_{center}} + \zeta \underbrace{\left\| \sum_{i=1}^{N} \left( \frac{h_{ci} - w^*}{\|h_{ci} - w^*\|} + \frac{h_{li} - w^*}{\|h_{li} - w^*\|} \right) \right\|^2}_{\mathcal{L}_{geomed}} - \eta \underbrace{\sum_{j=1}^{C} \sum_{k=j+1}^{C} \|w_j^* - w_k^*\|^2}_{\mathcal{L}_{separate}} \tag{7}$$

where $C$ denotes the number of FI nodes (e.g., the number of classes). The resulting loss function will be used to jointly train feature aligner $h$ and feature impression $w^*$.

## D   TRAINING DETAILS

The training process consists of two stages: the first stage involves training the baseline model from scratch using a proper training set, ensuring the pre-trained model has no knowledge of the calibration instances. The second stage involves training the feature aligner and feature impression nodes.

For the first training stage, the baseline models were trained using the same training objective and hyperparameters (e.g., number of epochs, learning rate, etc.) as those used in existing models. In the second training stage, a single batch size was used with the same hyperparameteres as the first stage. Here, we use the proposed training loss function (Eq. 7). We set $\delta = \frac{5}{\text{num\_queries}}$, $\zeta = \frac{3}{\text{num\_queries}}$, and $\eta = \frac{1}{7 \times \text{num\_queries}}$. The coefficient values are determined empirically for stable loss convergence and improved learning of positions. For $\delta$ and $\zeta$, we observed that $\mathcal{L}_{center}$ and $\mathcal{L}_{geomed}$ values scale similarly, but $\mathcal{L}_{center}$ converges more slowly. Therefore, we assigned a higher priority to $\delta$ than $\zeta$ with a ratio of 5:3.

Regarding $\eta$, if the contribution of $\mathcal{L}_{separate}$ is small, the distance between $w^*$ of different classes becomes smaller, causing ambiguities in distinguishing between them. Conversely, a large contribution of $\mathcal{L}_{separate}$ disrupts loss convergence and training stability. Through iterative adjustments, we found that $\frac{1}{7 \times \text{num\_queries}}$ yields optimal results. Fig. 8 shows the loss variation with respect to the $\eta$ value.

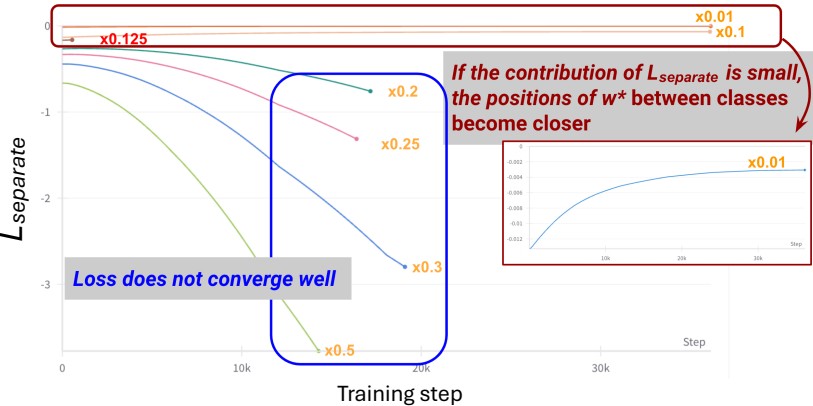

Figure 8: Coefficient Values in Training Objective. This plot shows the effect of the gamma value on $L_{separate}$ with seven separate training progressions. Different curves represent different training settings of gamma values. The text located next to each curve indicates the gamma value for each training ($\frac{1}{num\_queries}$ is omitted). This figure shows how we can choose the gamma value, which ensures both better loss convergence and stable learning of $w^*$.

## E    ABLATION STUDY

This section presents an ablation study to assess the contribution of each component. As the feature aligner (FA) and feature impression (FI) are designed for uncertainty quantification based on conformal prediction, we conducted the ablation study hierarchically as follows.

**Uncertainty quantification (UQ) vs. Object-level dynamic weighting.** Table 7 demonstrates that object-level dynamic weighting is ineffective without UQ, and vice versa, highlighting their mutual dependence. When both are disabled, static fusion is performed through concatenation. Without conformal prediction, a simple MLP model generates a weight value for each object query. When object-level dynamic weighting is disabled, the nonconformity score in UQ is computed for the entire input feature.

Table 7: Performance evaluation of different configurations under various conditions. ✓ indicates the presence of a component.

| UQ | Dynamic Weighting | No Corruption | Rainy Day | Clear Night | Rainy Night | Cam. Blackout | Avg. Latency (sec) |
|---|---|---|---|---|---|---|---|
|  |  | 66.16 | 68.34 | 44.51 | 27.26 | 45.13 | 0.91 ± 0.02 |
|  | ✓ | 63.01 | 63.51 | 42.01 | 21.91 | 37.01 | 0.91 ± 0.03 |
| ✓ |  | 66.2 | 68.37 | 44.52 | 27.3 | 45.21 | 1.26 ± 0.04 |
| ✓ | ✓ | 66.8 | 68.89 | 45.68 | 27.98 | 51.87 | 1.27 ± 0.04 |

**Uncertainty Quantification Analysis.** We also conduct an ablation study within the scope of uncertainty quantification, examining: 1) The impact of feature aligner (FA) layer count (Table 8), 2) The impact of feature impression (FI) vector dimension (Table 9), 3) The impact of the portion of decoder layers involved in UQ (Table 10). By integrating these findings, we determine the best configuration combination (Table 11). Applying this configuration across all decoder layers reduces the overhead to **0.24 seconds**, while improving accuracy by **0.58%** over static fusion.

Table 8: Performance evaluation with varying FA layer counts.

| FA layer count | FI dimension | No Corruption | Rainy Day | Clear Night | Rainy Night | Cam Blackout | Avg. Latency (sec) |
|---|---|---|---|---|---|---|---|
| 2 | 128 | 66.01 | 68.18 | 44.39 | 26.99 | 43.89 | 1.16 ± 0.05 |
| 4 | 128 | 66.67 | 68.56 | 45.53 | 27.41 | 50.99 | 1.19 ± 0.04 |
| 6 | 128 | 66.82 | 68.90 | 45.55 | 27.56 | 51.45 | 1.24 ± 0.05 |
| 8 | 128 | 66.80 | 68.89 | 45.68 | 27.98 | 51.87 | 1.27 ± 0.04 |

Table 9: Performance evaluation with varying FI dimensions.

| FA layer count | FI dimension | No Corruption | Rainy Day | Clear Night | Rainy Night | Cam Blackout | Avg. Latency (sec) |
|---|---|---|---|---|---|---|---|
| 8 | 32 | 66.73 | 68.91 | 45.62 | 27.91 | 51.81 | $1.16 \pm 0.03$ |
| 8 | 64 | 66.72 | 68.96 | 45.64 | 27.85 | 51.79 | $1.21 \pm 0.04$ |
| 8 | 128 | 66.80 | 68.89 | 45.68 | 27.98 | 51.87 | $1.27 \pm 0.04$ |

Table 10: Performance evaluation with varying UQ decoder counts.

| UQ Decoder Count | FA Layer Count | FI Dimension | No Corruption | Rainy Day | Clear Night | Rainy Night | Cam Blackout | Avg. Latency (sec) |
|---|---|---|---|---|---|---|---|---|
| 3 | 8 | 128 | 66.28 | 68.17 | 45.11 | 27.32 | 51.3 | $1.13 \pm 0.03$ |
| 6 | 8 | 128 | 66.80 | 68.89 | 45.68 | 27.98 | 51.87 | $1.27 \pm 0.04$ |

# F    EVALUATION ON TRANSFUSION

To demonstrate Cocoon's applicability to other transformer-based detectors, we evaluate it using TransFusion (Bai et al., 2022). TransFusion consists of 2D (ResNet50 He et al. (2016) and FPN Lin et al. (2017)) and 3D (VoxelNet Zhou & Tuzel (2018)) encoders, along with a single transformer decoder layer that uses an *element-wise sum* to combine the 2D and 3D features obtained from the attention mechanisms. See Appendix G for further reconfiguration details.

As shown in Table 12, Cocoon achieves better or comparable accuracy under various corruption scenarios compared to baseline methods. These results use a small set of the nuScenes validation set (120 samples). Since TransFusion has only a single transformer decoder layer, we cannot apply Cal-DETR (T) and Cal-DETR (Q) (Munir et al., 2023), which require multiple decoder layers for uncertainty quantification.

# G    BASE MODEL RECONFIGURATION DETAILS

**FUTR3D.** To use FUTR3D as our base model, concatenation (for feature fusion) is replaced with an element-wise sum. This modification maintains accuracy, with both configurations achieving an mAP of 67.39% after training with the proper training set.

**TransFusion.** TransFusion consists of 2D and 3D encoders, along with a single transformer decoder layer that uses an element-wise sum to combine the features obtained from attention mechanism. Therefore, unlike FUTR3D, TransFusion satisfies our requirement for an element-wise sum.

Currently, due to the need for calibration data separate from the training data, we train the model using only the proper training set. However, obtaining an independent calibration set (100 samples) would allow us to skip the first training stage, requiring only the feature aligner and FIs to be trained with the original nuScenes training set.

# H    OFFLINE PREPARATION FOR CONFORMAL PREDICTION

During the offline stage of conformal prediction preparation, we first prepare a proper training set and calibration set by splitting the original training set, unless a separately collected dataset is available for calibration. The pivotal process is the calibration step, which computes nonconformity scores (see Eq. 1) for each instance in the calibration set, establishing benchmarks against which new test instances are evaluated.

**Model Reconfiguration.** Depending on the initial fusion approach used, specific adjustments are made to the model. For instance, models initially using feature concatenation are reconfigured to adopt an element-wise sum, supplemented by a simple multi-layer perceptron (MLP) with one or two fully connected layers, allowing for a linear combination at a 1:1 fusion ratio. This reconfiguration is confirmed to maintain detection accuracy, as exemplified by the FUTR3D model, which retains its original accuracy of 67.3% post-adaptation. Additional details on model reconfiguration are discussed in Appendix G.

**Training.** This phase involves retraining the model with the designated training set, utilizing the specialized loss function detailed in Sec. 4.4. The focus is on aligning and imprinting features to ensure the model's readiness for conformal prediction.

Table 11: Best configuration combination for uncertainty quantification.

| UQ decoder count | FA layer count | FI dimension | No Corruption | Rainy Day | Clear Night | Rainy Night | Cam Blackout | Avg. Latency (sec) |
|---|---|---|---|---|---|---|---|---|
| 6 | 4 | 32 | 66.59 | 68.21 | 45.17 | 27.08 | 50.45 | $1.15 \pm 0.03$ |

Table 12: Accuracy (mAP) gain of Cocoon across various corruptions and object categories. 'Ours': the integration of TransFusion and Cocoon. Categories that do not belong to any scene are omitted. Base$_{Trans}$: (Bai et al., 2022), AdapNet*: (Valada et al., 2017)

| Corruption Type | Baseline | Car | Truck | Bus | Pedestrian | Bicycle | T.C. | Total |
|---|---|---|---|---|---|---|---|---|
| No Corruption | **Base**$_{Trans}$ | **88.2** | 43.7 | 93.4 | 95.4 | 57.5 | 65.5 | 74.0 |
| | AdapNet* | 86.7 (-1.5) | **52.4** (+8.7) | **94.3** (+0.9) | 94.9 (-0.5) | 54.7 (-2.8) | 60.4 (-5.1) | 73.9 (-0.1) |
| | **Ours** | **88.2** (+0) | 45.1 (+1.4) | 93.6 (+0.2) | **95.6** (+0.2) | **59.3** (+1.8) | **66.6** (+1.1) | **74.7** (+0.7) |
| Misalignment | **Base**$_{Trans}$ | **86.7** | **34.9** | 92.4 | 94.1 | **42.5** | 61.2 | **68.6** |
| | AdapNet* | 85.1 (-1.6) | 16.9 (-18.0) | **93.1** (+0.7) | 93.3 (-0.8) | 40.6 (-1.9) | 54.8 (-6.4) | 64.0 (-4.6) |
| | **Ours** | 86.4 (-0.3) | 33.4 (-1.5) | 92.1 (-0.3) | **94.3** (+0.2) | **42.5** (+0) | **62.1** (+0.9) | 68.5 (-0.1) |
| Blackout (C) | **Base**$_{Trans}$ | 84.7 | 27.0 | **94.7** | 93.8 | **43.3** | 59.6 | 67.2 |
| | AdapNet* | 83.8 (-0.9) | 24.5 (-2.5) | 92.5 (-2.2) | 91.3 (-2.5) | 41.1 (-2.2) | 54.9 (-4.7) | 64.7 (-2.5) |
| | **Ours** | 84.6 (-0.1) | **30.4** (+3.4) | **94.7** (+0) | **93.9** (+0.1) | 40.2 (-3.1) | **59.9** (+0.3) | **67.3** (+0.1) |
| Sampling (L) | **Base**$_{Trans}$ | **86.6** | 41.4 | 91.4 | **94.7** | 54.0 | 66.6 | 72.5 |
| | AdapNet* | 84.6 (-2.0) | **50.7** (+9.3) | **91.9** (+0.5) | 93.8 (-0.9) | 54.1 (+0.1) | 61.6 (-5.0) | 72.8 (+0.3) |
| | **Ours** | **86.6** (+0) | 44.3 (+2.9) | 91.4 (+0) | **94.7** (+0) | **55.9** (+1.9) | **67.9** (+1.3) | **73.5** (+1.0) |

**Calibration.** This crucial step involves calculating nonconformity scores for all instances in the calibration set. These scores are derived from the nonconformity function: $NC_i = \|(h \circ f)(x_i) - w^*\|$.

Through these methodical subprocesses—model reconfiguration, training, and calibration—our framework thoroughly prepares the model for effective conformal prediction, ensuring it can generate accurate and reliable predictions consistent with the principles of conformal prediction.

## I  BASELINE ADAPTIVE METHODS.

Our study investigates various adaptive fusion methods, which serve as baseline approaches in our comparative analysis (see Table. 3). Since none of the existing methods were originally designed for 3D object detection, we adapt their fusion mechanisms to work with FUTR3D and train them using FUTR3D's original hyperparameters. This ensures that all baseline methods, along with Cocoon, share a common base model, allowing us to focus on comparing the effectiveness of the adaptive fusion mechanisms.

**AdapNet*** (Valada et al., 2017) employs a mixture-of-experts (MoE) approach with a trainable gating network to enable adaptive fusion, as illustrated in Fig. 1a. The gating network, placed after the 2D and 3D encoders, determines the feature weights for each modality. This mechanism learns scalar weights for both encoded features and applies the weights to the entire feature space of each modality, producing a fused feature. **Zhu et al. (2020a)** uses self-attention to determine the weights for each encoded feature. In this approach, the encoded features pass through a self-attention module, and a weighted sum is used to produce the fused feature. **Zhang et al. (2023)** employs a late fusion approach that introduces adaptiveness between detection outputs generated from separate detection pipelines. For the detection outputs of both modalities, they perform an informative data selection process using Non-Maximum Suppression (NMS). The decision-level fusion is achieved by combining the box proposals in an enhanced NMS process that incorporates both Intersection over Union (IoU) and confidence scores. In our implementation, we use two versions of the FUTR3D model—one for 2D data only and the other for 3D data only—and apply the NMS process to the box proposals from each modality. **Cal-DETR (Munir et al., 2023)** measures uncertainty in transformer-based perception architectures by calculating variance across the transformer decoder layers. Since this algorithm was originally designed for unimodal perception, we modified FUTR3D's architecture by adding two transformer blocks (two sets of decoder layers): one for 2D data and another for 3D data. These blocks independently assess the uncertainty for each modality. After applying the weighted sum, the adaptively fused input passes through the original transformer decoder to produce the output.

**Cal-DETR (T)** determines the weights at the tensor level, while **Cal-DETR (Q)** determines them at the query level, enabling us to evaluate the importance of object-level adaptive fusion.

## J    INSIGHTS FROM OUR NONCONFORMITY MEASURE ANALYSIS

In Sec. 4.3, we present the findings, also shown below, from applying our nonconformity measures to a real-world dataset (Caesar et al., 2019).

---

*Finding 1:* Feature uncertainty significantly affects the NC score of the top-1 FI node.

*Finding 2:* Higher feature uncertainty in transformers with multiple decoder layers leads to significant instability in the top-1 FI node across layers.

---

These observations were made using the FUTR3D model, which comprises six transformer decoder layers. Finding 1 is also noted with the TransFusion model. However, due to the TransFusion model having only one decoder layer, Finding 2 is not observed. We will elaborate on the findings using Figs. 9 and 10, and Tables 14, 15, and 16.

When measuring the nonconformity scores for multi-modal features, if the object is clearly visible (Fig. 9), the FI nodes corresponding to the two modalities are mostly the same. The nonconformity scores for the top-1 FI node and the second most promising FI node show a significant difference (Table 14). For Object 1, the correct label corresponds to the car class (FI node **0**), and both sensors predict this accurately, facilitating equal fusion between the camera and LiDAR.

However, when the object is unclear (e.g., overlapping or occlusion cases) (Fig. 10), the nonconformity score for the top-1 FI node is very high, with no significant difference from the second most promising FI node (Table 15). For Object 2, the correct label is the construction vehicle class (FI node **2**), but the FI node selection for both modalities does not match this class. Nevertheless, the LiDAR modality predicted a more stable FI node with a lower top-1 nonconformity score, indicating more certain information.

For Object 3 (a person next to a traffic cone and construction vehicle), recognition with LiDAR is challenging. Here, the camera, excelling at color and small object detection, leads to better perception performance, aligning with our nonconformity measure in terms of top-1 FI node consistency and nonconformity scores (Table 16). Object 3's correct label is the pedestrian class (FI node **8**), and the camera predicts this more stably with a lower nonconformity score compared to the LiDAR. Therefore, the camera is deemed more certain in this case.

| FI Node # | 0 | 1 | 2 | 3 | 4 | 5 | 6 | 7 | 8 | 9 |
|---|---|---|---|---|---|---|---|---|---|---|
| Class | Car | Truck | C.V. | Bus | Trailer | Barrier | M.C. | Bicycle | Ped | T.C. |

Table 13: FI Node Clarification. In this experiment, FI nodes are designed to indicate classes. 'C.V.': Construction Vehicle, 'M.C.': Motorcycle, 'T.C.': Traffic Cone.

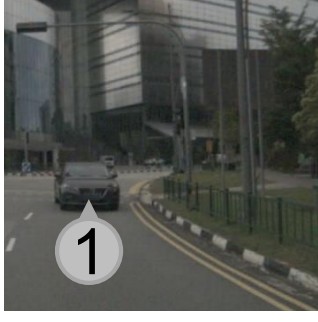

Figure 9: Clearly Visible Car.

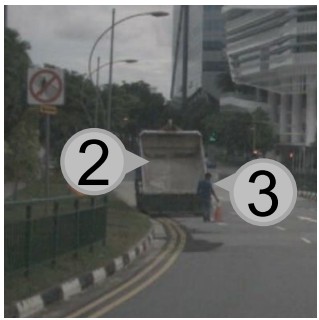

Figure 10: Challenging Case.

**(a) Top-3 NC Scores Sorted in Ascending Order**

| Object 2 | Camera NC Scores | | | LiDAR NC Scores | | |
|---|---|---|---|---|---|---|
| Layer | 1 | 2 | 3 | 1 | 2 | 3 |
| 0 | 0.08 | 9.08 | 9.75 | 0.09 | 9.24 | 9.88 |
| 1 | 0.07 | 10.14 | 11.06 | 0.09 | 10.23 | 11.17 |
| 2 | 0.06 | 10.65 | 12.22 | 0.06 | 10.74 | 12.31 |
| 3 | 0.06 | 11.00 | 11.24 | 0.07 | 11.11 | 11.35 |
| 4 | 0.06 | 12.66 | 12.81 | 0.08 | 12.75 | 12.90 |
| 5 | 0.07 | 13.56 | 13.70 | 0.07 | 13.66 | 13.80 |

**(b) Corresponding FI Nodes For Top-3 NC Scores**

| Object 2 | Camera FI Nodes | | | LiDAR FI Nodes | | |
|---|---|---|---|---|---|---|
| Layer | 1 | 2 | 3 | 1 | 2 | 3 |
| 0 | 0 | 4 | 1 | 0 | 4 | 1 |
| 1 | 0 | 2 | 1 | 0 | 2 | 1 |
| 2 | 0 | 1 | 2 | 0 | 1 | 2 |
| 3 | 0 | 1 | 9 | 0 | 1 | 9 |
| 4 | 0 | 5 | 8 | 0 | 5 | 8 |
| 5 | 0 | 1 | 2 | 0 | 1 | 2 |

Table 14: NC Score Analysis for Object 1. Correct class in blue.

**(a) Top-3 NC Scores Sorted in Ascending Order**

| Object 2 | Camera NC Scores | | | LiDAR NC Scores | | |
|---|---|---|---|---|---|---|
| Layer | 1 | 2 | 3 | 1 | 2 | 3 |
| 0 | 1.31 | 8.26 | 9.11 | 3.82 | 6.05 | 6.73 |
| 1 | 3.91 | 10.74 | 11.68 | 2.40 | 6.25 | 7.49 |
| 2 | 7.09 | 21.22 | 21.41 | 4.79 | 5.98 | 6.00 |
| 3 | 0.21 | 6.92 | 8.69 | 0.34 | 6.79 | 8.60 |
| 4 | 2.16 | 18.21 | 18.98 | 0.20 | 12.94 | 13.00 |
| 5 | 6.26 | 10.44 | 12.26 | 4.61 | 7.90 | 10.16 |

**(b) Corresponding FI Nodes For Top-3 NC Scores**

| Object 2 | Camera FI Nodes | | | LiDAR FI Nodes | | |
|---|---|---|---|---|---|---|
| Layer | 1 | 2 | 3 | 1 | 2 | 3 |
| 0 | 8 | 9 | 4 | 4 | 9 | 1 |
| 1 | 2 | 0 | 4 | 4 | 9 | 1 |
| 2 | 3 | 2 | 1 | 1 | 2 | 4 |
| 3 | 1 | 4 | 2 | 1 | 4 | 2 |
| 4 | 3 | 1 | 9 | 1 | 9 | 0 |
| 5 | 1 | 3 | 4 | 4 | 2 | 1 |

Table 15: NC Score Analysis for Object 2. Correct class in blue.

**(a) Top-3 NC Scores Sorted in Ascending Order**

| Object 3 | Camera NC Scores | | | LiDAR NC Scores | | |
|---|---|---|---|---|---|---|
| Layer | 1 | 2 | 3 | 1 | 2 | 3 |
| 0 | 0.07 | 9.28 | 10.28 | 0.24 | 9.50 | 10.54 |
| 1 | 0.12 | 12.57 | 12.60 | 5.19 | 11.61 | 13.22 |
| 2 | 0.10 | 11.31 | 12.11 | 0.22 | 11.57 | 12.36 |
| 3 | 0.09 | 10.75 | 12.72 | 0.16 | 10.98 | 12.92 |
| 4 | 0.07 | 12.56 | 12.80 | 8.71 | 10.23 | 12.44 |
| 5 | 0.08 | 12.23 | 12.59 | 0.33 | 13.02 | 13.20 |

**(b) Corresponding FI Nodes For Top-3 NC Scores**

| Object 3 | Camera FI Nodes | | | LiDAR FI Nodes | | |
|---|---|---|---|---|---|---|
| Layer | 1 | 2 | 3 | 1 | 2 | 3 |
| 0 | 8 | 9 | 4 | 8 | 9 | 4 |
| 1 | 8 | 0 | 9 | 6 | 0 | 8 |
| 2 | 8 | 4 | 9 | 8 | 4 | 9 |
| 3 | 8 | 4 | 9 | 8 | 4 | 9 |
| 4 | 8 | 9 | 0 | 8 | 6 | 0 |
| 5 | 8 | 4 | 1 | 9 | 1 | 8 |

Table 16: NC Score Analysis for Object 3. Correct class in blue.

## K RUNTIME OVERHEAD

| | Static: Base model | Adaptive: Base model + Cocoon |
|---|---|---|
| **FUTR3D** | $0.91 \pm 0.02$ | $1.27 \pm 0.04$ |
| **TransFusion** | $0.7 \pm 0.1$ | $0.7 \pm 0.1$ |

Table 17: Comparison of inference latency (in seconds) between static and adaptive fusion. Run 120 times for mean and standard deviation.

Table 17 shows the runtime overhead when Cocoon is integrated. In TransFusion, which uses a single transformer decoder layer processing 200 queries, there is no significant increase in latency compared to static fusion. However, in FUTR3D, which has six transformer decoder layers processing 900 queries, the average latency increases by 0.5 seconds, mainly due to computing nonconformity scores in all six layers. To minimize this latency increase, we performed nonconformity computation only on the last three layers. In this case, the latency can be reduced to $1.3 \pm 0.1$ seconds, achieving 66.28% accuracy in normal (no corruption) condition and 51.3% accuracy in 2D data corruption (blackout) condition. To further reduce the latency gap between static fusion and Cocoon's adaptive fusion, we can use 200 queries instead of 900, as in TransFusion, or apply model compression (e.g., quantization and pruning) to the feature aligners.

## L FINE-TUNING-BASED COMPUTATIONAL OVERHEAD MITIGATION

From the ablation study E, we found that adjusting the MLP layer count and the vector dimension of Feature Impression (FI) can reduce the overhead of 120 ms while maintaining higher accuracy than static fusion.

To further reduce the overhead, we devised a fine-tuning approach: dynamic fusion is applied only to the last decoder layer, and we fine-tune only the components in the later stages of the fusion process (the earlier parts must remain unchanged to preserve the validity of nonconformity score).

Using this fine-tuning approach, we reduced Cocoon's additional overhead (excluding the base model) to 50 ms—a significant reduction of 86% from the previous overhead of 360 ms—while achieving the final accuracy shown in Table 18 on an NVIDIA GeForce RTX 2080 (11 TOPS; Burnes (2019)). Given the computational capability of current AV systems —the most common field where Cocoon can be applied (254 TOPS; NVIDIA)—this overhead is negligible, with total latency within the 100 ms threshold for real-time guarantees (Lin et al., 2018).

Table 18: Performance comparison under different conditions.

| Fusion | No Corruption | Rainy Day | Clear Night | Rainy Night | Point Sampling (L) | Random Noise (C) | Avg. Latency (sec) |
|---|---|---|---|---|---|---|---|
| Static | 66.16 | 68.34 | 44.51 | 27.26 | 65.17 | 60.39 | $0.91 \pm 0.02$ |
| Cocoon (original) | 66.80 | 68.89 | 45.68 | 27.98 | 65.89 | 61.83 | $1.27 \pm 0.04$ |
| Cocoon (fine-tuning) | 67.13 | 69.12 | 45.70 | 28.07 | 66.02 | 61.99 | $0.96 \pm 0.03$ |

## M   HANDLING LARGE QUALITY DIFFERENCES ACROSS MODALITIES

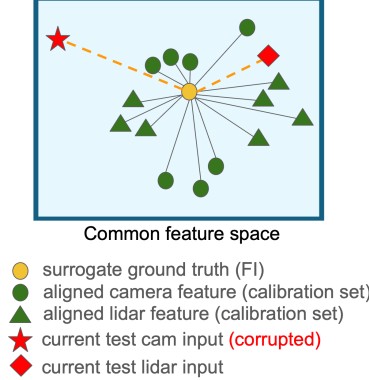

Figure 11: Feature Alignment Outcomes

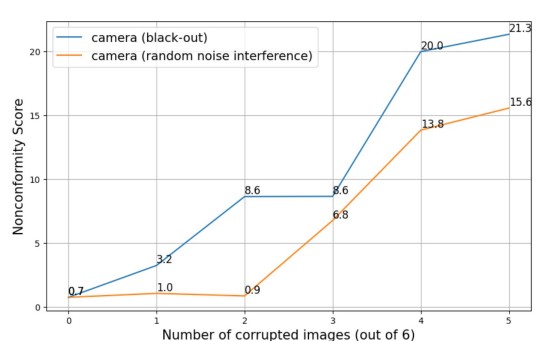

Figure 12: Nonconformity Scores Under Varying Camera Corruption Levels

Figs. 11 and 12 show the feature alignment outcomes and nonconformity scores under conditions of large quality differences across modalities. In this scenario, low-quality data points are positioned farther from the learned surrogate ground truth (i.e., Feature Impression), resulting in lower assigned weights.

## N   LICENSE OF ASSETS

**Code.** FUTR3D[2] and TransFusion[3] are both under the Apache-2.0 license.

**Datasets.** we utilize the nuScenes dataset (Caesar et al., 2019), which is licensed under CC BY-NC-SA 4.0. Appendix 5.3 shows ten different datasets. Several of these datasets, including Bio, Bike, Community, Facebook1, and Facebook2, are sourced from the UCI Machine Learning Repository (Kelly et al., 2023; Singh, 2016), which is licensed under CC BY 4.0. The Blog dataset (Buza, 2014) is also licensed under CC BY 4.0, while the Star dataset (Achilles et al., 2008) is under the CC0 1.0 license. For the Medical Expenditure Panel Survey datasets (Meps19, Meps20, and Meps21) (Cohen et al., 2009), we have the necessary permissions for their use, as we utilize these datasets exclusively for statistical reporting and analysis (Agency for Healthcare Research and Quality, 2017).

---

[2]https://github.com/Tsinghua-MARS-Lab/futr3d
[3]https://github.com/XuyangBai/TransFusion

