# OpenReview forum: "Cocoon: Robust Multi-Modal Perception with Uncertainty-Aware Sensor Fusion"
_ICLR.cc/2025/Conference — ICLR 2025 Poster_

### Official Review · Reviewer_38S7 · 2024-10-30

**Soundness:** 3
**Presentation:** 4
**Contribution:** 3
**Rating:** 6
**Confidence:** 3

**Summary:**

T​this paper propose a multimodal fusion framework called Cocoon for 3D object detection. The core innovation of Cocoon is to achieve object and feature level fusion through uncertainty quantification. The framework introduces a feature aligner and a learnable alternative truth value (called "Feature Impression") to fairly compare uncertainties between multimodalities and dynamically adjust the weight of each mode. Experimental results show that Cocoon outperforms existing static and adaptive fusion methods under standard and challenging conditions, and performs more robustly under natural and artificial perturbations.

**Strengths:**

1. The paper is clearly presented and easy to understand.
2. The innovation of uncertainty quantification mechanism is feasible from the principles and experiments in Section 4.
3. What I care about more is that the Cocoon framework has good scalability and can be further applied to more types of modal fusion, such as combining camera, LiDAR and radar data for autonomous driving, or combining visual and language modalities for visual language models (VLM).

**Weaknesses:**

1. The entire COCOON process, especially the structural comparison shown in Figure 1, and the "FEATURE IMPRESSION-BASED FEATURE ALIGNER" and "UNCERTAINTY QUANTIFICATION" parts in Sec 4.2 and 4.3, made me feel the huge computational overhead. Whether it is based on RGB, LiDAR, or multi-modal fusion target detection methods, I think they cannot be implemented quickly if the real-time performance cannot be guaranteed. The advantages can only be reflected at the numerical level of the paper.
2. Does COCOON rely too much on the quality of feature alignment and calibration in the data? Section 4.2 mentioned that since data from different modalities need to be projected into the same feature space for comparison, the accuracy of this part of the alignment directly affects the reliability of uncertainty quantification. If there are large differences in the quality or sampling frequency of the features of each modality, it will be difficult for the feature aligner to effectively align the data distribution, will this in turn affect the overall fusion effect?
3. At the end of Section 5.1, the value of the hyperparameter $ \alpha $ is set, but Table 4 directly assigns $ \alpha $ to 0.1. And Limitations mentions that the change of alpha will bring about a significant change in the algorithm effect. Is COCOON too dependent on hyperparameters?

**Questions:**

See Weaknesses.

---

> ### Author Response · Authors · 2024-11-17
> **Authors' Response (1/2): Computational Overhead and Clarification**
>
> Dear reviewer 38S7, we genuinely thank you for highlighting the issues regarding feasibility (in terms of computational overhead), the confusing presentation in the previous version, and areas requiring further clarification. Please find our responses addressing these concerns.
>
> ---
> **Q1: I think they cannot be implemented quickly if the real-time performance cannot be guaranteed. The advantages can only be reflected at the numerical level of the paper.**
>
> **A1:** Please refer to General Response 1.
>
> To summarize, the overhead of our uncertainty quantification primarily arises from the feature alignment process. In this process, features pass through a pretrained simple MLP (8 layers), without requiring the iterative search used in the reference work Feature CP [1]. This approach significantly reduces the time overhead, enabling us to complete uncertainty quantification with merely 50 ms overhead on an NVIDIA GeForce RTX 2080 (11 TOPS). Given the computational capability of current AV systems (254 TOPS), this overhead is negligible, with total latency within the 100 ms threshold for real-time guarantees.  More details on overhead analysis are provided in General Response 1.
>
> ---
> **Q2:  If there are large differences in the quality or sampling frequency of the features of each modality, it will be difficult for the feature aligner to effectively align the data distribution, will this in turn affect the overall fusion effect?**
>
> **A2:** With the feature aligner and the entire detector trained on the same training set, the feature aligner provides a clue about which modality is more beneficial for the detector to maximize its decoding capability. Thanks to this clue, Cocoon’s adaptive fusion outperforms static fusion, achieving higher accuracy in scenarios with large disparity in feature quality across modalities. Below is how Cocoon operates under significant quality differences across modalities.
>
> The disparity in feature quality can usually be attributed to (1) environmental noise (e.g., a scenario like nighttime where one modality’s data quality is significantly reduced) or (2) sensor corruption (e.g., a situation where one sensor fails). Both cases belong to the long-tail category, the key focus in this work. Thus, Cocoon handles these situations more effectively than standard fusion techniques.
>
> - For (1) environmental noise, please see the results of natural corruptions in Table 2.
> - For (2) sensor corruption, please see the results of artificial corruptions in Table 2.
>
> To summarize, when there is a disparity in feature quality, Cocoon assigns higher weights to high-quality features, thus leading to better perception performance. This is achieved through the feature aligner’s clue, which helps determine the optimal fusion ratio by identifying the degree of long-tail characteristics (relative to the training data) at the fusion point, prior to the detector’s decoding process.
>
> For clarity, we have included additional visualizations in ***Appendix K***, showing feature alignment outcomes (Fig. 11) and nonconformity scores (Fig. 12) under these conditions. As illustrated, low-quality data points are positioned farther from the learned surrogate ground truth (i.e., Feature Impression), leading them to receive lower weights.
>
> In the meantime, the difference in sampling frequency actually falls outside our scope. Like most 3D sensor fusion studies, we assume pre-processed (synchronized) data pairs (e.g., camera and LiDAR) as input. Although the camera and LiDAR may operate at different native frequencies (e.g., camera captured at 12 FPS and lidar 20 FPS in nuScenes dataset [2]), pre-processing ensures synchronization. Therefore, our focus is on the perception module with synchronization assumed, which is consistent with other recent multimodal 3D detection works [3-7].
>
> ---
> ***Reference***
>
> [1] Teng, Jiaye, et al. "Predictive inference with feature conformal prediction." ICLR 2023.
>
> [2] https://www.nuscenes.org/nuscenes?tutorial=maps
>
> [3] Xie, Yichen, et al. "Sparsefusion: Fusing multi-modal sparse representations for multi-sensor 3d object detection." ICCV. 2023.
>
> [4] Chen, Xuanyao, et al. "Futr3d: A unified sensor fusion framework for 3d detection." CVPRW. 2023.
>
> [5] Li, Yanwei, et al. "Unifying voxel-based representation with transformer for 3d object detection." NeurIPS 2022
>
> [6] Yang, Zeyu, et al. "Deepinteraction: 3d object detection via modality interaction." NeurIPS 2022.
>
> [7] Chen, Zehui, et al. "Deformable feature aggregation for dynamic multi-modal 3D object detection." ECCV 2022

---

> > ### Author Response · Authors · 2024-11-17
> > **Authors' Response (2/2): Clarification**
> >
> > **Q3:** At the end of Section 5.1, the value of the hyperparameter  is set, but Table 4 directly assigns  to 0.1. And Limitations mentions that the change of alpha will bring about a significant change in the algorithm effect. Is COCOON too dependent on hyperparameters?
> >
> > **A3:** I apologize for any confusion caused by the repeated use of the symbol alpha ($\alpha$) and appreciate you pointing this out. Each $\alpha$ previously had a distinct meaning, which we have clarified in the revised PDF (with modified text marked in blue). In the updated version, $\alpha$ exclusively represents the weight of one modality’s features, as shown in Fig. 3, while $\beta$ represents the weight of the other modality’s features. **Importantly, our $\alpha$ is automatically determined without relying on hyperparameters.**
> >
> > Below is the meaning of each alpha:
> >
> > > 1. At the end of Section 5.1, the value of the hyperparameter alpha is set (==> Eq. 3 and Line 462)
> >
> > >> This $\alpha$ represents the coefficient of the first term in our training loss function (Equation 3). To avoid confusion, we changed this symbol to $\delta$ (and also replaced previous $\beta$ and $\gamma$ with $\zeta$ and $\eta$, respectively) in Eq. 3 and Line 462. The derivation of how this value is obtained is shown in Appendix D.
> >
> > > 2. but Table 4 directly assigns alpha to 0.1 (==> Section 4.1, Line 518, and Table 4)
> >
> > >> This $\alpha$ is the significance level used in conformal prediction, introduced in Section 4.1. We have changed all occurrences of this $\alpha$ to $\Lambda$. This parameter is not directly related to Cocoon but is used to explain and validate conformal prediction. In Section 4.1, this significance level is used to explain conformal prediction; in Section 5.3, it validates the reliability of our uncertainty quantification (absolute difference calculation).
> >
> > > 3. And Limitations mentions that the change of alpha will bring about a significant change in the algorithm effect. (==> all other sections)
> >
> > >> This $\alpha$ denotes the weight of one modality determined through uncertainty quantification, as shown in Fig. 3. We retain the $\alpha$ and $\beta$ notation to indicate each modality’s weight, which is automatically determined by the Cocoon.

---

> ### Author Response · Authors · 2024-11-23
>
> Dear Reviewer 38S7,
>
> We hope our responses have adequately addressed your concerns about computational overhead and clarified any other points. If you have additional questions or concerns, please let us know, and we would be happy to provide further clarification. We truly appreciate your feedback and look forward to hearing from you soon.
>
> Best regards,
> Authors

---

> > ### Comment · Reviewer_38S7 · 2024-11-25
> > **Authors' Response Reply**
> >
> > The author's response answered my questions about the paper well.
> > ***
> > The symbolic issues in this paper are confusing, and I hope the author will check them further.

---

> ### Author Response · Authors · 2024-11-25
> **Response to Reviewer 38S7**
>
> First of all, we sincerely appreciate your reply and thank you for raising the score to the positive side!
>
> Regarding your comment, "_The symbolic issues in this paper are confusing, and I hope the author will check them further_," we acknowledge that our paper introduces many symbols, which may affect readability. We apologize for the confusion caused in the first version, where most of these were referred to as $\alpha$. To improve clarity, we included a symbol summary table (shown below) in Section 4 of the paper.
>
> | Symbol         | Meaning                                                                                                                                       | Purpose                  | Section Containing This Symbol         |
> |----------------|-----------------------------------------------------------------------------------------------------------------------------------------------|--------------------------|----------------------------------------|
> | $\alpha$, $\beta$    | Key symbols; $\alpha$ represents the weight of one modality (e.g., camera), and $\beta$ represents the weight of another (e.g., LiDAR). These weights are dynamically determined by the Cocoon mechanism through uncertainty quantification. | Uncertainty-based weights (Dynamic)    | Fig. 3, Fig. 6, Fig. 7, Sec. 4.3, Sec. 5.2             |
> | $\delta$, $\zeta$, $\eta$ | Coefficients in the training loss function (Equation 3).                                                                                     | Hyperparameter (Fixed)        | Sec. 5.1, Eq. 3                        |
> | $\Lambda$         | Significance value used solely to explain the concept of conformal prediction.                                                                       | Analysis (Fixed)                | Sec. 4.1, Sec. 5.3, Table 4                 |
>
> == Below is added  ==
>
> We have uploaded a revised version to alleviate any inconvenience for readers. Below are the modifications made in the revision:
>
> * Added Table 2 to summarize the symbols presented in our paper. Since our paper introduces many symbols, this table helps readers easily look up their meanings and purposes.
>
> * Revised Section 4.1 through 4.4 to make sentences clearer and more concise, improving readability.
>
> * Updated Section 4.3 with additional justification for the uncertainty quantification formula, along with reference.
>
> * Included Appendix K to clarify how our feature aligner operates and how the nonconformity function values change in the presence of noise or corruption in the data.
>
> We hope these revisions address any potential confusion and make the paper easier to understand. Thank you again for the time and effort you’ve put into reviewing our work!

---

### Official Review · Reviewer_9JUv · 2024-11-03

**Soundness:** 3
**Presentation:** 3
**Contribution:** 2
**Rating:** 6
**Confidence:** 4

**Summary:**

The paper introduces Cocoon, a novel framework that addresses the challenge of multi-modal fusion in 3D object detection. The paper proposes a novel uncertainty-aware fusion framework that operates at both object and feature levels through the introduction of Feature Impression and a feature alignment mechanism. This approach leverages conformal prediction theory to quantify uncertainties in heterogeneous sensor data, enabling dynamic weight adjustment between different modalities such as cameras and LiDAR.

**Strengths:**

1.Proposes a novel framework (Cocoon) that uniquely combines object-level and feature-level uncertainty awareness in multi-modal fusion, addressing a significant gap in existing approaches.
2.Thoroughly evaluates the model under diverse corruption scenarios, including both natural corruptions.
3.Provides solid mathematical foundation for the uncertainty quantification method, with clear derivations and validations across multiple datasets.

**Weaknesses:**

1.The paper shows a substantial increase in computational latency - from 1.1s to 1.6s when integrating Cocoon with FUTR3D.
2.While the paper demonstrates consistent improvements over baselines, the actual gains are relatively modest in many scenarios.
3.Only evaluates camera and LiDAR fusion, missing other important sensors like radar and ultrasonic.
4. The lack of validation on other datasets raises concerns about the generalizability of the method. Validation can be performed on the Waymo and Argoverse 2 datasets.
5. The method provides marginal gains and performs worse than previous methods such as CMT, BEVFusion, and DeepInteraction.

**Questions:**

1.While you propose using fewer decoder layers or queries to reduce latency: How does this affect the accuracy-speed tradeoff specifically?
2.Could you provide ablation studies on the importance of different components in your framework? Such as, how much does each component (feature aligner, Feature Impression, uncertainty quantification) contribute to the final performance?

---

> ### Author Response · Authors · 2024-11-17
> **Authors' Response (1/2): Computational Overhead, Ablation Study, and Generalizability**
>
> Dear reviewer 9JUv, we greatly appreciate you bringing up the issues regarding feasibility (in terms of computational overhead), lack of ablation study, and generalizability. Please find our responses and additional experimental results addressing these concerns.
>
> ---
> **Q1:** Computational Overhead
>
> **A1:** First, regarding the use of fewer decoder layers for uncertainty quantification, please refer to General Response 2, where we present the results as an ablation study. However, we found that this approach might be too model-specific to generalize across other potential base models.
>
> Instead, we developed a fine-tuning-based solution that significantly reduces computational overhead while improving accuracy. **Please see General Response 1 for further details.**
>
> ---
> **Q2:**  Lack of Ablation Study
>
> **A2:** see General Response 2
>
> ---
> **Q3:** The method provides marginal gains and performs worse than previous methods such as CMT, BEVFusion, and DeepInteraction.
>
> **A3:** There are two reasons for the lower accuracy:
>
> The first reason lies in the difference in the training set. As noted in Appendix A, we trained the detector on a partial training set and used the remaining frames for the calibration stage. If we had separately obtained calibration instances (approximately 100 samples) without splitting the training set, we could have utilized the entire training set to train the detector. When CMT is trained on the same partial training set, it achieves 66.83%, demonstrating that Cocoon enables our base model, FUTR3D, to outperform CMT, as shown below.
>
>
> |               | No Corruption | Rainy Day | Clear Night | Rainy Night | Point Sampling (L) |Random Noise (C) |
> |---------------|-------------|----------|-------------|------------|-------------|-----------|
> | **CMT [1]**       | 66.83       | 68.44    | 44.63       | 27.31      | 65.08       | 60.35     |
> | FUTR3D (basemodel)  | 66.16       | 68.34    | 44.51       | 27.26      | 65.17       | 60.39     |
> | FUTR3D + Cocoon (original) | 66.80       | 68.89    | 45.68       | 27.98      | 65.89       | 60.56     |
> | FUTR3D + Cocoon (fine-tuning) | 67.13  | 69.12    | 45.70       | 28.07      | 66.02       | 61.99     |
>
> Second, please note that our algorithm focuses on fusion rather than building the entire perception model. Since the different components of each model hinder direct comparison, we implemented various fusion methods on FUTR3D and TransFusion to isolate and evaluate the effectiveness of the fusion method. When compared with other fusion methods—Concatenation (used by CMT [1] and BEVFusion [2]) and Attention (used by DeepInteraction [3])—Cocoon achieves the highest accuracy, as shown in Table 2.
>
> ---
> **Q4:**  Only evaluates camera and LiDAR fusion, missing other important sensors like radar and ultrasonic.
>
> **A4:** Thank you for raising this issue. In this work, we primarily focus on the most representative modalities (lidar and camera). Given the similar data types of LiDAR and radar (both being point clouds), Cocoon can be easily applied to camera-radar fusion. As noted in Sec. 6, we are currently extending our research to include additional modalities, particularly for VLM.

---

> > ### Comment · Reviewer_9JUv · 2024-11-27
> > **Accuracy**
> >
> > Thank you for your detailed response. While we appreciate Cocoon's improvement in handling complex scenarios, we hope it maintains its base performance as well. For the nuScenes dataset, could you provide a table showing performance under the official setting, particularly focusing on the nuScenes Detection Score (NDS), as it is a key metric?

---

> ### Author Response · Authors · 2024-11-21
> **Authors' Response (2/2): Generalizability**
>
> ---
> **Q5:**  The lack of validation on other datasets raises concerns about the generalizability of the method. Validation can be performed on the Waymo and Argoverse 2 datasets.
>
> **A5:** nuScenes is widely recognized as a comprehensive real-world driving dataset. It covers diverse cities, times, and weather conditions, providing sufficient diversity to prove Cocoon’s generality. In fact, most recent works [4–8] exclusively demonstrate their impact using the nuScenes dataset. Building on this, we also focus on the nuScenes dataset, integrating various types of artificial and natural noise. We are producing results on the Waymo dataset to be included in future versions.
>
> Given that the most remarkable distinction between the Waymo dataset and Argoverse2 is the LiDAR beam density—nuScenes uses a 32-beam LiDAR, Waymo uses a 64-beam LiDAR, and Argoverse2 uses a 128-beam LiDAR—point cloud data exhibits different levels of sparsity. Based on this, for an additional generalizability check, we experiment with different LiDAR beam densities. The results below demonstrate that Cocoon outperforms static fusion across varying LiDAR data sparsity levels.
>
>
> | Beam Type | Fusion  | No Corrupt | Point Sampling (L) | Random Noise (C) | Sensor Misalignment |
> |-----------|---------|------------|--------------------|-------------------|---------------------|
> | 32-beam   | Static  | 66.16      | 65.17             | 60.39            | 53.16              |
> |           | Cocoon  | 66.80      | 65.89             | 61.83            | 55.61              |
> | 16-beam   | Static  | 60.20      | 59.17             | 54.50            | 47.17              |
> |           | Cocoon  | 60.87      | 59.97             | 55.91            | 49.59              |
> | 4-beam    | Static  | 55.80      | 54.74             | 50.01            | 42.34              |
> |           | Cocoon  | 56.62      | 55.69             | 51.63            | 45.32              |
>
> ---
>
> ***Reference***
>
> [1] Yan, Junjie, et al. "Cross modal transformer: Towards fast and robust 3d object detection." Proceedings of the IEEE/CVF International Conference on Computer Vision. 2023.
>
> [2] Liu, Zhijian, et al. "Bevfusion: Multi-task multi-sensor fusion with unified bird's-eye view representation." 2023 IEEE international conference on robotics and automation (ICRA). IEEE, 2023.
>
> [3] Yang, Zeyu, et al. "Deepinteraction: 3d object detection via modality interaction." Advances in Neural Information Processing Systems 35 (2022): 1992-2005.
>
> [4] Xie, Yichen, et al. "Sparsefusion: Fusing multi-modal sparse representations for multi-sensor 3d object detection." ICCV. 2023.
>
> [5] Chen, Xuanyao, et al. "Futr3d: A unified sensor fusion framework for 3d detection." proceedings of the IEEE/CVF conference on computer vision and pattern recognition. 2023.
>
> [6] Li, Yanwei, et al. "Unifying voxel-based representation with transformer for 3d object detection." Advances in Neural Information Processing Systems 35 (2022): 18442-18455.
>
> [7] Yang, Zeyu, et al. "Deepinteraction: 3d object detection via modality interaction." Advances in Neural Information Processing Systems 35 (2022): 1992-2005.
>
> [8] Chen, Zehui, et al. "Deformable feature aggregation for dynamic multi-modal 3D object detection." European conference on computer vision. Cham: Springer Nature Switzerland, 2022.

---

> ### Author Response · Authors · 2024-11-23
>
> Dear Reviewer 9JUv,
>
> We hope that our responses have adequately addressed your concerns regarding computational overhead, the lack of an ablation study, and Cocoon’s generalizability. If you have additional questions or concerns, please let us know, and we would be happy to provide further clarification. We truly appreciate your feedback and look forward to hearing from you soon.
>
> Best regards,
> Authors

---

> ### Author Response · Authors · 2024-11-27
> **Response to Reviewer 9JUv**
>
> Thank you for your comments and for raising your score to the positive side!
>
> Regarding your additional questions, we would like to seek clarification on your comments, particularly concerning `base performance` and `official setting`.
>
> **[1] While we appreciate Cocoon's improvement in handling complex scenarios, we hope it maintains its `base performance` as well.**
>
> >By `base performance`, we assume you are referring to performance on the standard nuScenes dataset without artificial or natural noise. As shown in the `No Corruption` column of our Table 2, Cocoon outperforms the baseline methods in all normal and challenging scenarios. Additionally, the accuracy breakdown (in Table 3) demonstrates that Cocoon surpasses the baseline in normal setups, regardless of object configuration (distance & size). Could you clarify which aspect of the base performance you believe was not adequately maintained?
>
> ---
>
> **[2] For the nuScenes dataset, could you provide a table showing performance under the `official setting`, particularly focusing on the nuScenes Detection Score (NDS), as it is a key metric?**
>
> > If we understand correctly, by `official setting`, you are referring to the evaluation results (e.g., NDS, class-wise AP, Errors) obtained using the nuScenes-devkit API.
> If this is correct, pease find the table below ($\uparrow$: higher is better, $\downarrow$: lower is better).
>
> | Scenario          | Fusion Method | mAP $\uparrow$ | mATE $\downarrow$ | mASE $\downarrow$ | mAOE $\downarrow$ | mAVE $\downarrow$ | mAAE $\downarrow$ | NDS $\uparrow$ | Car_AP $\uparrow$ | Bike_AP $\uparrow$ | Motorcycle_AP $\uparrow$ | Truck_AP $\uparrow$ |
> |-------------------|---------------|---------|----------|----------|----------|----------|----------|---------|------------|---------|------|-------|
> | No Corruption     | Static        | 66.16   | 33.10    | 26.13    | 26.60    | 30.92    | 19.09    | 69.49   | 86.4       | 63.5    | 74.9 | 61.8  |
> |                   | Cal-DETR      | 66.24   | 32.83    | 26.07    | 26.56    | 30.43    | 19.10    | 69.62   | 86.4       | 63.5    | 74.6 | 62.0  |
> |                   | **Cocoon**        | **66.80**   | 32.94    | 25.92    | 26.40    | 29.87    | 19.20    | **69.97**   | 86.8       | 65.1    | 75.9 | 62.0  |
> | Rainy Day         | Static        | 68.34   | 31.68    | 27.56    | 21.56    | 21.51    | 13.43    | 72.59   | 88.6       | 65.6    | 80.6 | 64.3  |
> |                   | Cal-DETR      | 68.37   | 31.50    | 27.41    | 21.70    | 21.31    | 13.85    | 72.64   | 88.6       | 65.8    | 80.9 | 64.3  |
> |                   | **Cocoon**        | **68.89**   | 31.53    | 27.23    | 21.68    | 21.00    | 13.88    | **72.92**   | 88.9       | 66.1    | 82.1 | 64.3  |
> | Clear Night       | Static        | 44.51   | 50.09    | 46.37    | 43.59    | 63.29    | 58.24    | 46.10   | 87.7       | 54.3    | 72.5 | 80.4  |
> |                   | Cal-DETR      | 44.38   | 50.02    | 46.50    | 44.18    | 62.78    | 57.78    | 46.21   | 87.8       | 54.5    | 71.9 | 80.1  |
> |                   | **Cocoon**        | **45.68**   | 50.03    | 46.45    | 44.15    | 61.62    | 57.76    | **46.84**   | 88.4       | 59.9    | 75.5 | 82.0  |
> | Rainy Night       | Static        | 27.26   | 67.34    | 67.01    | 86.07    | 120.10   | 59.28    | **25.66**   | 88.5       | 0.0     | 71.7 | 81.8  |
> |                   | Cal-DETR      | 26.29   | 67.32    | 67.02    | 85.99    | 127.01   | 66.01    | 25.01   | 87.6       | 0.0     | 70.9 | 81.5  |
> |                   | **Cocoon**        | **27.98**   | 67.34    | 66.77    | 85.86    | 126.27   | 66.28    | 25.36   | 89.4       | 0.0     | 78.4 | 81.8  |
> | Point Sampling (L)| Static        | 65.17   | 33.58    | 26.17    | 27.40    | 31.67    | 19.08    |  68.80   | 85.6       | 61.8    | 72.2 | 61.2  |
> |                   | Cal-DETR      | 65.29   | 33.35    | 26.12    | 27.18    | 30.95    | 19.09    | 68.98   | 85.6       | 61.6    | 72.4 | 61.3  |
> |                   | **Cocoon**        | **65.89**   | 33.45    | 25.98    | 26.92    | 30.59    | 19.12    | **69.34**   | 86.0       | 63.3    | 73.4 | 61.5  |
> | Random Noise (C)  | Static        | 60.39   | 33.73    | 26.92    | 26.09    | 32.37    | 19.43    | 66.34   | 84.9       | 51.6    | 67.9 | 56.0  |
> |                   | Cal-DETR      | 60.40   | 33.37    | 26.76    | 25.66    | 31.86    | 19.42    | 66.50   | 84.9       | 51.7    | 68.0 | 56.2  |
> |                   | **Cocoon**       | **61.83**   | 33.42    | 26.58    | 25.38    | 31.12    | 19.47    | **67.32**   | 85.5       | 54.2    | 70.0 | 57.2  |
>
>
> Please let us know if our understanding is incorrect or if further clarification is needed. Thank you again for your comments!

---

> > ### Comment · Reviewer_9JUv · 2024-11-28
> >
> > Thank you for your positive response. Your understanding is correct.

---

### Official Review · Reviewer_XHYu · 2024-11-03

**Soundness:** 3
**Presentation:** 2
**Contribution:** 3
**Rating:** 6
**Confidence:** 4

**Summary:**

This paper introduces an object- and feature-level uncertainty-aware fusion framework for robust multi-modal sensor fusion in 3D detection scene of autonomous driving. It designs a valid uncertainty estimator for heterogeneous representations, and then dynamically fuses multi-modal sensor features. Experiments indicate the effectiveness of fusing heterogeneous features in several unrobust scene.

**Strengths:**

(1) It provides a very intuitive feature fusion analysis to derive the motivation of this paper
(2) All the designs that underpin the motivation of this article are very valid and reasonable.
(3) The results are rich and convincing.

**Weaknesses:**

Although the writing logic and thinking are clear, the explanation of the method is too complicated, which will bring inconvenience to the reader.

**Questions:**

No.

---

> ### Author Response · Authors · 2024-11-17
> **Authors' Response: Method Presentation**
>
> Dear reviewer XHYu, we truly appreciate you bringing up the issues in our algorithm presentation.
>
> We identified one factor that hinders readers’ understanding: the repetitive use of the same symbol ($\alpha$) caused some inconvenience, so we revised it to improve clarity and eliminate potential ambiguity. Please refer to the blue-colored text in the updated PDF.
>
> Additionally, we plan to add a video showing the overall mechanism (Figures 2 and 3) on our project website to minimize any potential confusion.
>
> If there are any specific parts you find overly complicated, we will revise them and share the updated version within this discussion session.
>
> Along with this, please refer to the general response for details on our further improvements and ablation study.

---

> ### Author Response · Authors · 2024-11-26
>
> Dear Reviewer,
>
> We have uploaded a revised version to alleviate any inconvenience for readers. Below are the modifications made in the revision:
>
> * Added Table 2 to summarize the symbols presented in our paper. Since our paper introduces many symbols, this table helps readers easily look up their meanings and purposes.
>
> * Revised Section 4.1 through 4.4 to make sentences clearer and more concise, improving readability.
>
> * Updated Section 4.3 with additional justification for the uncertainty quantification formula, along with reference.
>
> * Included Appendix K to clarify how our feature aligner operates and how the nonconformity function values change in the presence of noise or corruption in the data.
>
> While we received feedback from another reviewer that our paper is “easy to understand,” we acknowledge that some inconvenience may arise due to the complexity of the Cocoon mechanism, which involves multiple algorithms (e.g., conformal prediction, 3D object detection, Weiszfeld’s algorithm). We sincerely hope these revisions address any potential confusion and make the paper easier to understand.
>
> Since the revision upload period ends tomorrow, we kindly request you to review the updated version and further results in the General Response. Your attention to our modifications and responses is greatly appreciated. Thank you once again for the time and effort you've put into reviewing our work.

---

> > ### Comment · Area_Chair_UvwH · 2024-11-30
> >
> > Dear Reviewer,
> >
> > Thank you again for your efforts in reviewing this submission. It has been some time since the authors provided their feedback. We kindly encourage you to review their responses and other reviews, verify whether they address your concerns, and submit your final ratings. If you have additional comments, please initiate a discussion promptly. Your timely input is essential for progressing the review process.
> >
> > Best regards,
> >
> > AC

---

### Author Response · Authors · 2024-11-17
**General Response (1 & 2-1): Computational Overhead Mitigation & Ablation Study**

## [1] Computational Overhead Mitigation

First of all, we reduced inference latency through code refactoring: static fusion from 1.1 to 0.91 seconds and Cocoon from 1.6 to 1.27 seconds.

From the ablation study (described in next response), we found that adjusting the MLP layer count and the vector dimension of Feature Impression (FI) can reduce the overhead of 120 ms while maintaining higher accuracy than static fusion.

To further reduce the overhead, we devised a fine-tuning approach: dynamic fusion is applied only to the last decoder layer, and we fine-tune only the components in the later stages of the fusion process (the earlier parts must remain unchanged to preserve the validity of nonconformity score).

Using this fine-tuning approach, we reduced Cocoon’s additional overhead (excluding the base model) to 50 ms—**a significant reduction of 86% from the previous overhead value of 360 ms**—while achieving the final accuracy shown below on an NVIDIA GeForce RTX 2080 (11 TOPS [1]). **Given the computational capability of current AV systems (254 TOPS) [2], this overhead is negligible, with total latency within the 100 ms threshold for real-time guarantees [3].**

| Fusion                      | No Corruption | Rainy Day | Clear Night | Rainy Night | Point Sampling (L) | Random Noise (C) | Avg. Latency  (sec)     |
|----------------------------|---------------|-----------|-------------|------------|---------------------|------------------|--------------------|
| Static       | 66.16         | 68.34     | 44.51       | 27.26      | 65.17               | 60.39            | 0.91 ± 0.02       |
|  Cocoon (original) | 66.80         | 68.89     | 45.68       | 27.98      | 65.89               | 61.83           | 1.27 ± 0.04        |
| Cocoon (fine-tuning) | 67.13      | 69.12     | 45.70       | 28.07      | 66.02               | 61.99            | 0.96 ± 0.03        |

***Reference***


[1] https://www.nvidia.com/en-us/geforce/news/geforce-rtx-20-series-super-gpus/

[2] https://www.nvidia.com/en-us/self-driving-cars/in-vehicle-computing/

[3] Lin, Shih-Chieh, et al. "The architectural implications of autonomous driving: Constraints and acceleration." Proceedings of the Twenty-Third International Conference on Architectural Support for Programming Languages and Operating Systems. 2018.

---
## [2] Ablation Study

Given that feature aligner (FA) and feature impression (FI) are designed for uncertainty quantification (based on conformal prediction), we conducted the following experiments to evaluate each component's impact.

### [2-1]   Uncertainty quantification (UQ) vs. Object-level dynamic weighting

- When both are disabled, static fusion is performed via concatenation.
- When conformal prediction is disabled, a simple MLP model is used to output a weight value for each query.
- When dynamic weighting is disabled, the nonconformity (NC) score in UQ is measured for the entire input feature.

| Conformal Prediction-based UQ | Dynamic Weighting |  No Corruption | Rainy Day | Clear Night | Rainy Night  | Cam Blackout | Avg. Latency (sec)         |
| -------------------- | ----------------- | ----------- | -------- | ----------- | ---------- | -------- | ------------- |
|                |                  |  66.16         | 68.34     | 44.51       | 27.26      |  45.13            | 0.91 ± 0.02       |
|                | V                 | 63.01       | 63.51    | 42.01       | 21.91      | 37.01    | 0.91 $\pm$ 0.03   |
| V                    |            | 66.2        | 68.37    | 44.52       | 27.3       | 45.21    | 1.26 $\pm$ 0.04 |
| V                    | V                 | 66.8        | 68.89    | 45.68       | 27.98      | 51.87    | 1.27 $\pm$ 0.04 |


	Conclusion:  Dynamic weighting cannot function without UQ, and UQ is ineffective without dynamic weighting, making them both essential to each other.

---

> ### Author Response · Authors · 2024-11-17
> **General Response (2-2): Ablation Study & Our Message**
>
> ### [2-2] Within uncertainty quantification
>
> - Impact of feature aligner (FA) layer count
>
> | FA layer count | FI dimension | No Corruption | Rainy Day | Clear Night | Rainy Night | Cam Blackout | Avg. Latency (sec) |
> | -------------- | ------------------- | ------------- | --------- | ----------- | ----------- | ------------ | ------------------ |
> | 2              | 128                 | 66.01         | 68.18     | 44.39       | 26.99       | 43.89        | 1.16 $\pm$  0.05       |
> | 4              | 128                 | 66.67         | 68.56     | 45.53       | 27.41       | 50.99        | 1.19 $\pm$  0.04        |
> | 6              | 128                 | 66.82         | 68.9      | 45.55       | 27.56       | 51.45        | 1.24 $\pm$  0.05        |
> | 8              | 128                 | 66.8          | 68.89     | 45.68       | 27.98       | 51.87        | 1.27 $\pm$ 0.04      |
>
> - Impact of feature impression (FI) vector dimension
>
> | FA layer count | FI dimension | No Corruption | Rainy Day | Clear Night | Rainy Night | Cam Blackout | Avg. Latency (sec) |
> | -------------- | ------------ | ------------- | --------- | ----------- | ----------- | ------------ | ------------------ |
> | 8              | 32           | 66.73         | 68.91     | 45.62       | 27.91       | 51.81        | 1.16 $\pm$  0.03       |
> | 8              | 64           | 66.72         | 68.96     | 45.64       | 27.85       | 51.79        | 1.21 $\pm$  0.04        |
> | 8              | 128          | 66.8          | 68.89     | 45.68       | 27.98       | 51.87        | 1.27 $\pm$  0.04      |
>
> - Impact of the portion of decoder layers involved in UQ
>
> | UQ Decoder Count | FA Layer Count | FI Dimension | No Corruption | Rainy Day | Clear Night | Rainy Night | Cam Blackout | Avg. Latency (sec)          |
> |------------------|----------------|--------------|---------------|-----------|-------------|-------------|--------------|--------------|
> | 3                | 8              | 128          | 66.28         | 68.17     | 45.11       | 27.32       | 51.3         | 1.13 ± 0.03  |
> | 6                | 8              | 128          | 66.8          | 68.89     | 45.68       | 27.98       | 51.87        | 1.27 ± 0.04  |
>
> - Best Configuration Combination
>
> | UQ decoder count| FA layer count | FI dimension | No Corruption | Rainy Day | Clear Night | Rainy Night | Cam Blackout | Avg. Latency (sec)     |
> | --------| -------- | ------------ | ------------- | --------- | ----------- | ----------- | ------------ | ---------------------- |
> | 6        | 4        | 32           | 66.59         | 68.21     | 45.17       | 27.08       | 50.45        | 1.15 $\pm$ 0.03 |
>
>
> 	Conclusion: Based on the above experiments, we identified the optimal configuration to balance accuracy and latency. Applying this configuration to all decoder layers reduces the overhead to 0.24 seconds, while achieving a 0.58% improvement in accuracy over static fusion.
>
> ---
>
> ## Our Message
>
> We sincerely thank the reviewers for their insightful feedback and valuable suggestions. We have highlighted the improvements made based on their comments and emphasized our key strengths as recognized in their feedback.
>
> - We have updated our paper by reflecting all the comments from reviewers. The main points are as follows:
>   - Presentation that may confuse readers (see blue-colored text in the updated PDF).
>   - Computational overhead and improvements to reduce it (see General Response 1).
>   - Lack of ablation study (see General Response 2).
>
>
> * Below are the key strengths of our work as highlighted by reviewers:
>   - Motivation
>     - Obvious analysis driving the motivation (**Reviewer XHYu**)
>     - Very valid and reasonable progression from motivation to solution design (**Reviewer XHYu**)
>   - Methodology
>     - Reasonableness (**Reviewer XHYu**), novelty (**Reviewer 9JUv**), feasibility, and scalability (**Reviewer 38S7**) of the uncertainty-aware sensor fusion solution design
>     - Solid mathematical foundation for the uncertainty quantification method, with clear derivations (**Reviewers 9JUv, 38S7**)
>   - Evaluation
>     - Rich and convincing results (**Reviewer XHYu**)
>     - Thorough evaluation of Cocoon on diverse corruption scenarios, including both natural and artificial corruptions (**Reviewer 9JUv**)
>     - Soundness of Cocoon's uncertainty quantification method, validated across multiple datasets (**Reviewers 9JUv, 38S7**)
>   - Presentation
>     - Clear thinking and logical writing (**Reviewers XHYu, 38S7**) & Easy to understand (**Reviewer 38S7**)
>
> These invaluable comments and suggestions will greatly contribute to enhancing the quality of Cocoon. Thank you once again for your time and thoughtful input!

---

### Meta-Review · Area_Chair_UvwH · 2024-12-21

**Metareview:**

This paper proposes a new object- and feature-level uncertainty-aware multimodal fusion framework for 3D object detection tasks. The proposed framework adopts a feature aligner and a feature impression strategy to achieve uncertainty quantification for heterogeneous representations. The proposed approach is novel and effective and consistently outperforms existing static and adaptive methods. Reviewer concerns are about the complicated writing, increased computational latency, and some missing experiments and analysis. After rebuttal, most of these concerns were addressed, and two reviewers improved their score from 5 to 6. Finally, all reviewers achieve a consistent opinion of acceptance. After reading the paper and all the discussions, the AC confirmed the recommendation.

**Additional Comments On Reviewer Discussion:**

This paper received three reviews.
Reviewer XHYu initially provided a score of 6 and raised some minor concerns about the writing. After rebuttal, the reviewer did not provide further comments. The other two reviewers both initially suggested a score of 5. After rebuttal, they thought the concerns were well addressed and improved the score to 6. Therefore, all reviews are consistent that the paper is marginally above the acceptance threshold. The AC did not find new evidence to overturn the opinions and suggested to accept.

---

### Decision · Program_Chairs · 2025-01-22

Accept (Poster)